# Origin, trophic transfer and recycling of particulate organic matter in two upwelling bays of Humboldt Current System: Insights from compound-specific isotopic compositions of amino acids

**Benjamín Srain** [1,2]*, **Edgart Flores** [3], **Jorge Valdés**[1], **Andrés Camaño**[2]

**1** Laboratorio de Sedimentología y Paleoambientes (LASPAL), Instituto de Ciencias Naturales A. von Humboldt, Facultad de Ciencias del Mar y de Recursos Biológicos, Universidad de Antofagasta, Antofagasta, Chile, **2** Sistemas Socio Ecológicos SpA, Concepción, Chile, **3** Department of Geological Sciences and Institute of Artic and Alpine Research, University of Colorado Boulder, Boulder, Colorado, United States of America

* biosrain@gmail.com

## Abstract

The Chilean upwelling bays are highly productive ecosystems shaped by their interactions with the open ocean. Although significant knowledge exists regarding their hydrodynamic and ecological processes, the spatial dynamics of trophic transfer and heterotrophic resynthesis of organic matter remain insufficiently understood. To address these knowledge gaps, we conducted a compound-specific isotope analysis of amino acids (CSIAA) on suspended and sinking particulate organic matter from Mejillones and Antofagasta bays, two oceanographic environments characterized by contrasting hydrodynamic conditions and topographic orientations. In Mejillones Bay, the CSIAA trophic positions for metazoan ($1.7 \pm 0.5$) and protozoan ($2.3 \pm 0.3$) were significantly higher compared to those in Antofagasta Bay (metazoans: $1.3 \pm 0.6$; protozoans: $1.5 \pm 0.3$), highlighting protozoans as primary trophic vectors. MixSIAR analysis indicated that phytoplankton is a key source of particulate organic matter in both bays; however, Mejillones Bay exhibited a greater proportion of microbially degraded organic matter. Enhanced heterotrophic resynthesis in Mejillones Bay ($\Sigma V$: 1.9–2.5) was associated with lower oxygen levels, increased concentrations of $NO_2^-$, and heightened stratification of the water column. Additionally, depth-dependent variations in $\delta^{15}N$ for phenylalanine and threonine indicated a greater solubilization of particles, which contributed to a reduction in the export of particulate organic matter (averaging $9 \pm 2$ mg C/m²/d). These findings underscore the critical role of the intricate interactions between the bay's topographic features and the physical and biological processes that ultimately influence the cycling trajectories of particulate organic matter in upwelling bays.

**Data availability statement:** All relevant data are within the paper and its Supporting Information files.

**Funding:** This research was funded by "Asociación de Industriales de Mejillones" (CR 4800). The funders had no role in study design, data collection and analysis, decision to publish, or preparation of the manuscript.

**Competing interests:** The authors have declared that no competing interests exist.

## 1. Introduction

Coastal upwelling is the primary physical process driving high biological productivity in Eastern Boundary Upwelling Systems [1–3]. In Chile, the upwelling system is divided into three regions based on large-scale oceanographic patterns and annual upwelling regimes [4]. The northern upwelling region (18–30°S) experiences weak to intermittent upwelling year-round [5,6], while the central and southern regions exhibit a highly seasonal upwelling regime characterized by intense upwelling during the austral spring and summer and downwelling in autumn and winter [7].

The bays within these ecosystems are connected to the coastal ocean, exhibiting distinct patterns and processes primarily influenced by their interaction with the ocean. Thus, the term "upwelling bay" refers to these bays, which are physically driven and chemically and biologically fueled by coastal upwelling from the adjacent ocean [8].

In these bays, coastal topography, particularly headlands, significantly alters wind and current patterns, creating wind shadows and intensified winds that influence upwelling intensity and distribution. Complex interactions among shelf jets, local wind stress, and thermal gradients generate variable circulation and stratification patterns, which effectively retain upwelled waters and plankton within bays, promoting high primary and secondary productivity [9–11].

In Chilean upwelling ecosystems, phytoplankton serves as the primary source of particulate organic matter (POM), including both living cells and detrital material (e.g., dead organic matter and microbial biomass) [12]. The predominance of phytoplankton-derived POM establishes detrital food webs as significant components of nutrient cycling [6,13,14]. Consequently, POM is recognized as the main source of N, C, and other biologically significant elements in the ocean, supporting both pelagic and benthic ecosystems [15].

In pelagic marine environments, suspended and sinking particles comprise algae and detrital N, with amino acids representing the largest characterizable fraction of organic N exported to the ocean's interior [16,17]. The isotopic fractionation of C and N in amino acids offers valuable insights into key metabolic cycles, sources of organic matter, trophic transfers, and the processes by which heterotrophic microbes rework organic matter during decomposition and transformation [18–29].

The $\delta^{15}$N patterns in amino acids serve as powerful indicators for understanding trophic changes, primarily because organisms preferentially metabolize the lighter N isotope ($^{14}$N) [25,30,31]. This selectivity results in variations in N isotopic composition across diverse tissues and trophic levels. Trophic amino acids (Tr-AA), such as asparagine (Asp), glutamic acid (Glu), alanine (Ala), isoleucine (Ile), leucine (Leu), valine (Val), and proline (Pro), exhibit significant enrichment at each successive step in the food chain. In contrast, source amino acids (Src-AA), including serine (Ser), phenylalanine (Phe), lysine (Lys), and tyrosine (Tyr), display relatively stable $\delta^{15}$N values across trophic levels [25].

Each amino acid is synthesized through distinct biosynthetic pathways in both autotrophic and heterotrophic organisms. As a result, $\delta^{15}$N-amino acid patterns can serve as effective indicators of N sources and the mechanisms involved in organic

matter alteration [20]. Within this conceptual framework, δ15N-amino acid data have been utilized to track various processes in detrital organic matter, including estimates of trophic transfers [32,33], the degree of intracellular heterotrophic bacterial resynthesis [27], and extracellular heterotrophic bacterial degradation [34,35].

In this study, we analyzed the δ15N-amino acid signatures of suspended and sinking POM collected from Mejillones Bay (MB) and Antofagasta Bay (AB), two distinct oceanographic environments situated in the northern upwelling ecosystem of Chile. Our primary objective was to investigate the enrichment patterns of δ15N in amino acids to enhance our understanding of POM cycling in these upwelling bays. This research also aimed to evaluate local-scale spatial variations in the origins and sources of POM while investigating the processes of trophic transfer and heterotrophic resynthesis within the study area.

## 2. Methods

### 2.1. Study site

The Mejillones Peninsula (ca., 23°S – 70°W; Fig 1) is the most prominent tectonic geographical feature along the northern coast of Chile. It spans 50 km in length and 20 km in width, consisting of three morphological units: two north-south trending mountain ranges separated by a coastal plain [36]. This peninsula disrupts the linearity of the coastline, creating two bays with opposing orientations: MB, which faces north towards the equator, and AB, oriented southward [8,37,38].

This area is situated within the Humboldt Current System, one of the most productive ecosystems globally [39,40]. The oceanographic conditions in MB are shaped by various forcings operating on differing spatiotemporal scales. Coastal

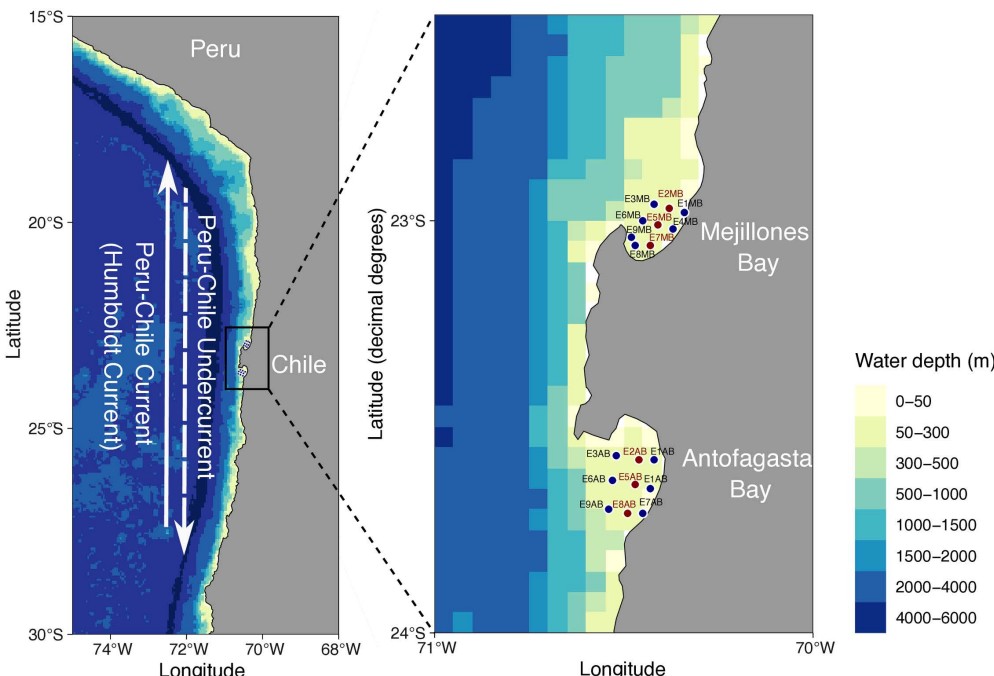

**Fig 1. Geographical context of the study area in northern Chile, shaped by the cold Humboldt Current flowing northward along the western coast of South America and the southward-moving Peru-Chile Undercurrent (left panel).** A detailed view of Mejillones and Antofagasta bays shows the nine sampling stations: blue circles indicate all general sampling locations, while red circles highlight the three stations in each bay where additional stable N and C isotope analyses were conducted (right panel). Reprinted from https://cran.r-project.org/web/packages/ggOceanMaps/index.html under a CC BY license, with permission from Dr. Edgart Flores, original copyright 2025.".

upwelling, driven by wind action, brings nutrient-rich but low-oxygen Subsurface Equatorial Water to the surface. This influx of water often results in hypoxic or suboxic conditions within the bay [6,37,41].

The bay's northern orientation influences the formation of upwelling shadows, largely determined by the coastal morphology. This phenomenon leads to the establishment of a thermal front at the mouth of the bay, which promotes stratification and retains water within the bay [8,36,37,41]. The interplay between vertical mixing and stratification processes in the water column, along with the presence of upwelling shadows, is essential for organism retention and fosters high primary productivity and organic matter generation within the bay [40,42].

AB is significantly influenced by the Humboldt Current System. The hydrodynamics within the bay are shaped by a thermal front generated by an upwelling center in the southern region, which regulates the bay's filling and emptying processes [10,38]. This dynamic model emphasizes that water enters the bay through the peninsular sector and exits through the southeastern sector, resulting in a cyclonic circulation pattern at the bay's center [4,43]. The formation of thermal fronts in AB acts as a physical barrier typical of upwelling zones, facilitating the retention of water masses near the coast—a phenomenon commonly referred to as the "upwelling trap" [8,38]. This trap effectively retains both organic and inorganic particles, as well as planktonic organisms [4]. The hydrodynamic conditions in AB are characterized by lower intensity surface currents compared to the adjacent oceanic zone. During the summer months, the dominant flow brings water into the bay, while in winter, the absence of winds often reverses this flow [4,8,38,43,44]. In contrast to MB, AB has been comparatively under-researched, with only a few studies examining its oceanographic characteristics, such as temperature variability patterns, seasonal dynamics of zooplankton, phytoplankton ecology, and the interplay between physical and biological processes within the bay.

## 2.2. Sampling

In March 2023, we conducted sampling at nine subtidal stations within MB and AB (Fig 1). Three sampling stations were established along the central axes of each bay for stable N and C isotope analyses (E2MB, E5MB, E7MB, E2AB, E5AB, and E8AB; see Fig 1).

Water samples were collected from four depths (5, 20, 35, and 45 m) using 10 L Niskin bottles, and were subsequently stored in acid-washed carboys in the dark at approximately 10 °C. In the laboratory, 30 L of seawater from each depth were filtered through pre-combusted 0.7 μm glass fiber filters (Whatman GF/F). The filtrates and filters were preserved at -80 °C until analysis, which included total organic carbon (TOC), chlorophyll-*a*, and nutrient concentrations ($NH_4^+$, $NO_2^-$, $NO_3^-$, and $PO_4^-$). Seawater samples for compound-specific $\delta^{15}N$ and $\delta^{13}C$ isotope analysis of amino acids in POM were collected from designated depths of 5, 20, 35, and 45 meters, along with additional samples for $\delta^{15}NO_3^-$ analysis. For the $\delta^{15}N\text{-}NO_3^-$ analysis, water was collected in 20 mL acid-washed plastic scintillation vials using a syringe fitted with a GF/F filter.

## 2.3. Hydrographic profiles

Continuous profiles of temperature (°C), salinity, dissolved oxygen (μM), and pH were measured using a CTD-O (SBE-19) instrument. All sampling and oceanographic activities conducted during the research cruises were authorized by the Chilean Navy's Hydrographic and Oceanographic Service (SHOA), in accordance with Ordinary SHOA Resolution.

## 2.4. Sediment traps deployment

During the cruise, surface-tethered sediment traps were deployed at the stations E5MB, E7MB, E2AB, E5AB, and E8AB (Fig 1) and submerged at a depth of 45 m for a duration of 30 days to collect sinking POM. The deployment system consisted of stainless-steel cross supports equipped with PVC pipe collectors and particle interceptor traps (PITs). Following the Joint Global Ocean Flux Study (JGOFS) protocol, each PIT was filled with a brine solution containing $HgCl_2$ to inhibit biological activity. Upon retrieval, the samples were subjected to filtration, freeze-drying, and weighing before being stored for subsequent particulate organic carbon (POC) analysis.

 

## 2.5. Preparation of filter samples

Filters and filtrates from water samples and sediment traps were examined under a dissecting microscope (10 to 40x magnification) to identify and remove contaminants, including swimmers, intact zooplankton, fibers, and plastic fragments. All samples were lyophilized for 24 hours at a pressure of approximately 0.05 mbar and a temperature of -80°C until all water was removed. Subsequently, the samples were stored at -20°C for the remainder of the processing.

## 2.6. Chemical analyses

### 2.6.1. Chlorophyl-*a* and nutrients.
For chlorophyll-*a*, 1 L samples of water were filtered (GF/F glass fiber filters) in triplicate and immediately frozen (-20 °C) until later. Chlorophyll-*a* concentrations were determined by fluorometry using a Shimadzu 1201 spectrophotometer after 20–24 h extraction into 90% acetone at 5°C following to Strickland and Parsons [45]. The dissolved inorganic nutrients $NH_4^+$, $NO_2^-$, $NO_3^-$, and $PO_4^{3-}$, were quantified using UV-VIS spectrophotometry, following the methodologies outlined by Strickland and Parsons [45] and Grasshoff et al. [46].

### 2.6.2. Total organic carbon.
For TOC analysis, the seawater samples were homogenized and diluted. A small aliquot was then injected into a heated reaction chamber containing a platinum catalyst, which facilitated the vaporization of water and the oxidation of organic carbon compounds to carbon dioxide and water. The resulting carbon dioxide from the oxidation process was quantified using nondispersive infrared detection with an elemental analyzer.

### 2.6.3. Particulate organic carbon.
For the analysis of POC, 1 L samples from sediment trap collectors were homogenized and filtered through pre-combusted GF/F fiberglass filters (0.7 μm) and stored at freezing temperatures until analysis. The POC content was determined using a Carlo Erba C/N analyzer, with acetanilide as the standard [47].

## 2.7. Ancillary oceanographic parameters

Dissolved inorganic N anomalies (N*), defined as a linear combination of $NO_3^-$ and $PO_4^{3-}$, were calculated to assess the roles and distribution of nitrogen fixation and denitrification in the water column. The N* parameter was estimated according to the method described by Hansell et al. [48], using the formula: $N^* = (NO_3^- + NO_2^- + NH_4^+) - 16 (PO_4^{3-}) + 2.9$. Values of N* lower than $-3$ μmol kg$^{-1}$ indicate denitrification, while values above 2 μmol kg$^{-1}$ suggest nitrogen fixation [49].

The apparent oxygen utilization (AOU), which indicates the amount of oxygen respired in the ocean interior, was calculated as follows:

$$AOU = (O_{2\ saturation} - O_{2\ measured})$$

Oxygen saturation as a function of temperature and salinity in the water column was determined according to Weiss RF. [50].

## 2.8. Dissolved oxygen inventories and water column stability

Oxygen inventories in the water column were calculated by integrating the vertical profiles of dissolved oxygen using the trapezoidal method in Sigma Plot 12.0 software. The stability of the water column was assessed by calculating the Brunt-Väisälä Frequency with Ocean Data View 4.0 software [51].

## 2.9. Isotopic analysis

### 2.9.1. δ$^{15}$N of dissolved $NO_3^-$.
The stable isotopic composition of naturally occurring N in $NO_3^-$ was analyzed using the denitrification method [52,53]. Prior to analysis, samples were treated with sulfamic acid to reduce and eliminate naturally occurring $NO_2^-$ [54,55]. The δ$^{15}$N values were measured in nitrous oxide ($N_2O$) that was quantitatively produced by denitrifying bacteria devoid of $N_2O$-reductase activity. The generated $N_2O$ was analyzed using gas chromatography coupled with continuous flow isotopic ratio mass spectrometry (GC CF-IRMS) (Finnigan Delta Plus).

Isotopic data are presented in delta notation as $\delta^{15}N = [(R(sample)/R(reference)) - 1] \times 1000$, where R represents the ratio of $^{15}N$ to $^{14}N$. The isotopic reproducibility was better than 0.2%. Samples were calibrated against the IAEA-NO-3 and USGS34 NO$_3^-$ $\delta^{15}N$ isotopic reference materials, following the calibration procedure outlined by McIlvin and Casciotti [56].

**2.9.2. Amino acid hydrolysis and isotopic analysis.** Suspended and sinking POM from water column samples and sediment traps were collected on 47 mm diameter glass fiber filters with a pore size of 0.7 μm, which had been pre-combusted at 450°C. The filtrates were then hydrolyzed and derivatized following previously established protocols [27,57]. Dried samples were hydrolyzed under standard conditions (6 N HCl for 20 h at 110 °C), and the resulting hydrolysate purified using cation exchange chromatography (Dowex 50WX8–400 ion exchange resin) [58]. Isopropyl-TFA derivatives were prepared as described by Silfer et al. [59]. Derivatized samples were analyzed by a Thermo Trace 1310 gas chromatograph with IsolinkII/ConfloIV (reactor 1000 °C), coupled to a Thermo Delta V isotope ratio mass spectrometer.

Amino acids were separated for $\delta^{15}N$ analyses using a BPX5 column (60 m × 0.32 mm, 1 μm film thickness; SGE Analytical Science, Trajan, Austin, TX, USA) and for $\delta^{13}C$ analyses using a DB-5 column (50 m × 0.32 mm 0.52 μm film thickness; Agilent Technologies, Santa Clara, CA, USA). Under our analytical conditions, both $\delta^{15}N$ and $\delta^{13}C$ values could be reproducibly measured for Ala, aspartic acid + asparagine (Asp), glutamic acid + glutamine (Glu), Leu, Ile, Pr, Val, Gly, Lys, Ser, Phe, Tyr, and threonine (Thr) . The reproducibility of these amino acids was typically less than 1‰. The $\delta^{13}C$ values of the amino acids were calculated from the measured values of their derivatives, following the method established by Silfer et al. [59]. This process demonstrated a reproducibility of less than 1.5‰, with corrections applied based on an amino acid mixture standard, for which the isotopic values were independently determined using offline elemental analyzer analysis. The directly measured $\delta^{15}N$-amino acids values were also corrected based on bracketing external standards, as described in McCarthy et al. [57].

The injector temperature was 250 °C with a split He flow rate of 2 mL/min. The GC temperature program for N isotope analysis was: initial temp = 70 °C hold for 1 min; ramp 1 = 10 °C/min to 185 °C, hold for 2 min; ramp 2 = 2 °C/min to 200 °C, hold for 10 min; ramp 3 = 30 °C/min to 300 °C, hold for 6 min. The GC temperature program for C isotope analysis was initial temp = 75 °C hold for 2 min; ramp 1 = 4 °C/min to 90 °C, hold for 4 min; ramp 2 = 4 °C/min to 185 °C, hold for 5 min; ramp 3 = 10 °C/min to 250 °C, hold for 2 min; ramp 4 = 20 °C/min to 300 °C, hold for 5 min.

## 2.10. Amino acid grouping

According to the behavior of individual $\delta^{15}N$ values within food webs, amino acids can be classified into two groups [25]:

i) Tr-AA: Ala, Val, Leu, Ile, Pro, Asp and Glu are strongly enriched with trophic transfer.

ii) Src-AA: Ser, Phe, Lys, and Tyr, whose $\delta^{15}N$ values remain largely unaltered with each trophic shift.

Given that Phe is recognized as an abundant amino acid and is considered the most stable reference source amino acid in published culture experiments [25,27], we concentrated our analysis on the origins and sources of suspended and sinking POM in the waters of MB and AB by exclusively utilizing the $\delta^{15}N$-Phe values.

## 2.11. $\delta^{15}N$ amino acids parameters for trophic transfer, and resynthesis

Trophic transfer was evaluated by estimating the trophic position of POM. The trophic position of metazoans (TP$_{Metazoan}$) was determined using the method outlined by Chikaraishi et al. [32] and was calculated as follows:

$$TP\ \text{Metazoan} = \frac{(\delta15NGlu - \delta15NPhe - 3.4)}{(7.6 + 1)}$$

For protozoans, the trophic position (TP$_{Protozoan}$) was calculated based on the methods outlined by Décima et al. [33,60] and Décima and Landry [61]:

$$TP \text{ Protozoan} = \frac{(\delta15N\text{Ala} - \delta15N\text{Phe} - 3.2)}{(4.5 + 1)}$$

The parameter $\Sigma V$, which indicates total heterotrophic resynthesis is defined as the average deviation in the $\delta^{15}N$ values of the Tr-AA Ala, Val, Asp, Glu, Ile, Leu and Pro [62]. Thus:

$$\Sigma V = 1/n \ \Sigma \ Abs\left(\chi_{AA}\right)$$

Where:

"$\chi$" is the deviation of each of the trophic amino acid with respect to the average of all trophic amino acids.

"n" is the total number of Tr-AA used in the calculation.

In this study, we use the term "heterotrophic resynthesis" as the heterotrophic reworking of proteinaceous material, mediated by multiple processes carried out by planktonic heterotrophic organisms, such as bacteria, archaea, protists and zooplankton, including hydrolysis, uptake and *de novo* synthesis, salvage amino acid incorporation into new protein, as well as strict catabolism, as was defined by McCarthy et al. [27]. $\Sigma V$ index values less than 1 are considered non-degraded POM; values between 1 and 2 correspond to metazoan resynthesis, and values higher than 2 indicate microbial resynthesis [27].

## 2.12. $\delta^{15}N$ and $\delta^{13}C$ of total hydrolysable amino acids

The $\delta^{15}N$ and $\delta^{13}C$ of total hydrolysable amino acids (THAA) were determined by calculating the mole percent weighted sum of the $\delta^{15}N$-amino acids values as follows:

$\delta^{15}N$ THAA=$\Sigma(\delta^{15}N$ AA x Mol% AA) and $\delta^{13}C$ THAA=$\Sigma(\delta^{13}C$ AA x Mol% AA)

The $\delta^{15}N$ and $\delta^{13}C$ of THAA, values were operationally considered here as a proxy for total proteinaceous $\delta^{15}N$ and $\delta^{13}C$ values for suspended and sinking POM, assuming the proteinaceous $\delta^{15}N$ and $\delta^{13}C$ estimations, as the integration of all N and C-containing entities in our POM samples [29].

## 2.13. Phenylalanine-normalized $\delta^{15}N$ values of alanine and threonine for marine organic matter end-members

Values for $\delta^{15}N$ Phe-normalized $\delta^{15}N$-Ala and $\delta^{15}N$-Thr (calculated as $\delta^{15}N$-amino acid - $\delta^{15}N$-Phe) were obtained for various marine organic matter end-members from previously published studies (S1 Table in S1 Data) [34,35,57,63–67]. These values were then compared with those obtained in this study to assess the relative contributions of different organic matter end-members and to identify the most probable sources of POM at the study site.

## 2.14. Statistical tests and Bayesian mixing model

The assumptions of homogeneity of variances (Levene's test) and normality of variables (Shapiro-Wilk test) were not met. Consequently, we assessed significant differences in the dependent variables—trophic position and microbial heterotrophic resynthesis—within and between bays using nonparametric Wilcoxon and Mann-Whitney tests. Spearman rank correlation coefficients were calculated to examine correlations. All statistical analyses were performed using Statistica software, version 12.0. We also performed analysis of similarities (ANOSIM), based on Euclidean distances (9999 permutations), to assess the significance of differences in similarity between sample groups. ANOSIM is a non-metric multivariate statistical method that does not assume a specific data distribution (normality or equality of variances) and produces a test statistic (R) to determine the presence of differences among groups. This analysis compared Phe-normalized $\delta^{15}N$ values of Ala and Thr in organic matter end-members [67] with those of suspended and sinking POM collected during this study.

To analyze the contributions of various end-members—including zooplankton, phytoplankton, microbially-degraded organic matter (MDOM), and fecal pellets (FP)—to suspended and sinking POM in both MB and AB, we employed the

Bayesian mixing model MixSIAR (Version 3.2.0). This model assumes multivariate normality, accounting for potential co-variation in tracer values. Statistical analyses were conducted using the vegan package [68] and the ggplot2 package [69] in R.

## 3. Results

### 3.1. Hydrography

The vertical physical-chemical profiles of MB and AB exhibited similar patterns (Fig 2). Both bays featured a warmer, less saline, and well-oxygenated surface layer (0–5 m) with temperatures ranging from 13 to 21 °C, salinities between 33.5 and 34.5 PSU, and dissolved oxygen levels that fluctuated between 11–370 μM, indicating the occurrence of hypoxic and suboxic conditions at deeper waters. A distinct thermocline and oxycline were observed between depths of 5 m and 35 m (Fig 2). Alkaline conditions were present in the upper 12 m of the water column, with pH values ranging from 8 to 8.6, while pH decreased to 7.8 at depths greater than 20 m (Fig 2). The water column data suggested the dominance of Subtropical Surface Water (STSW) in both bays, alongside interactions with Subantarctic Water (SAW) and Equatorial Subsurface Water (ESSW) (Fig 2).

### 3.2. Chlorophyll-*a*, and nutrient contents

The vertical profiles of chlorophyll-*a* in both bays exhibited the highest concentrations at the surface, declining sharply with depth. In MB, chlorophyll-*a* concentrations ranged from 0.03 to 3.5 mg/m³, while in AB, they varied from 0.04 to 10.4 mg/m³ (S1A and B Figs in S1 Data). In MB, $NH_4^+$ averaged 0.8 μM, ranging from 0.1 to 2.7 μM, with the lowest content at 35 m depth at all stations (S1C Fig in S1 Data). $NO_2^-$ showed a minimum at 35 m and peaked at 20 m, averaging 0.5 μM with a maximum of 2.8 μM (S2A Fig in S1 Data). $NO_3^-$ concentrations ranged from 0.1 to 23 μM, peaking at 35 m (S2C Fig in S1 Data). $PO_4^{3-}$ showed a clear increase with depth, reaching up to 3.1 μM (S3A Fig in S1 Data).

In AB, $NH_4^+$ averaged 0.4 μM, ranging from 0.1 to 1.5 μM (S1D Fig in S1 Data). $NO_2^-$ fluctuated from 0.1 to 2.3 μM, averaging 0.3 μM (S2B Fig in S1 Data). $NO_3^-$ was lower at the surface (5 m; average 1 μM) and reached up to 17 μM at depths of 35–45 m (S2D Fig in S1 Data). $PO_4^{3-}$ showed a similar vertical pattern to MB, showing contents ranging from 0.5 to 3.7 μM and averaging 2.4 μM (S3B Fig in S1 Data).

### 3.3. Dissolved oxygen inventories contents and water column stratification

In MB, the depth integrated contents of dissolved oxygen ranged from 3394 to 7548 mM/m², with an average of 4665 ± 1331 mM/m² (S4A Fig in S1 Data). In contrast, AB exhibited oxygen inventories between 2477 and 7772 mM/m², averaging 5265 ± 1426 mM/m² (S4A Fig in S1 Data). Water column stability in MB was significantly higher than in AB, as determined by the Mann-Whitney Test ($p = 0.00002$). In MB, the Brunt-Väisälä frequency peaked at 216 ± 4 cycles/h (S4B Fig in S1 Data), with an average of 11 ± 14 cycles/h. In contrast, AB exhibited values ranging from 0.7 to 62 cycles/h, with a mean of 9 ± 11 cycles/h (S4C Fig in S1 Data). Significant differences were especially pronounced between the depth layers of 5–20 meters (Mann-Whitney test; $p = 0.001$; S4B and C Figs in S1 Data).

### 3.4. Total organic carbon contents

In MB, TOC content ranged from 0.8 to 2 mg/L, with an average of 1.1 ± 0.4 mg/L. The vertical distribution pattern showed higher TOC content in the first 20 m (S3C Fig in S1 Data). In AB, the average TOC content was significantly higher than in MB, averaging 1.6 ± 0.2 mg/L (Mann-Whitney test; $p = 0.03$), with values ranging from 0.7 ± 0.1 to 3.7 ± 0.8 mg/L (S3D Fig in S1 Data).

### 3.5. Downward flux of particulate organic carbon

In MB, after 30 days of deployment, the sediment traps located at the center of the bay (E5BM; Fig 1) collected 97 ± 12 mg/L of POC, while the collectors deployed at the southernmost station (E7BM; Fig 1) retained 135 ± 52 mg/L of

POC. In AB, the highest content of POC collected in the traps occurred at the center of the bay (E5BA; Fig 1), with a POC concentration of 156 ± 42 mg/L, followed by 120 mg ± 18 mg/L in the sediment trap located at the southernmost station of the bay (E8BA; Fig 1). Based on the aspect ratio of the trap array and the deployment time, a mean downward flux value of 9 ± 2 mg/m²/day of POC was estimated for MB. In AB, the estimated mean flux of POC was 11 ± 1 mg/m²/day (Table 1).

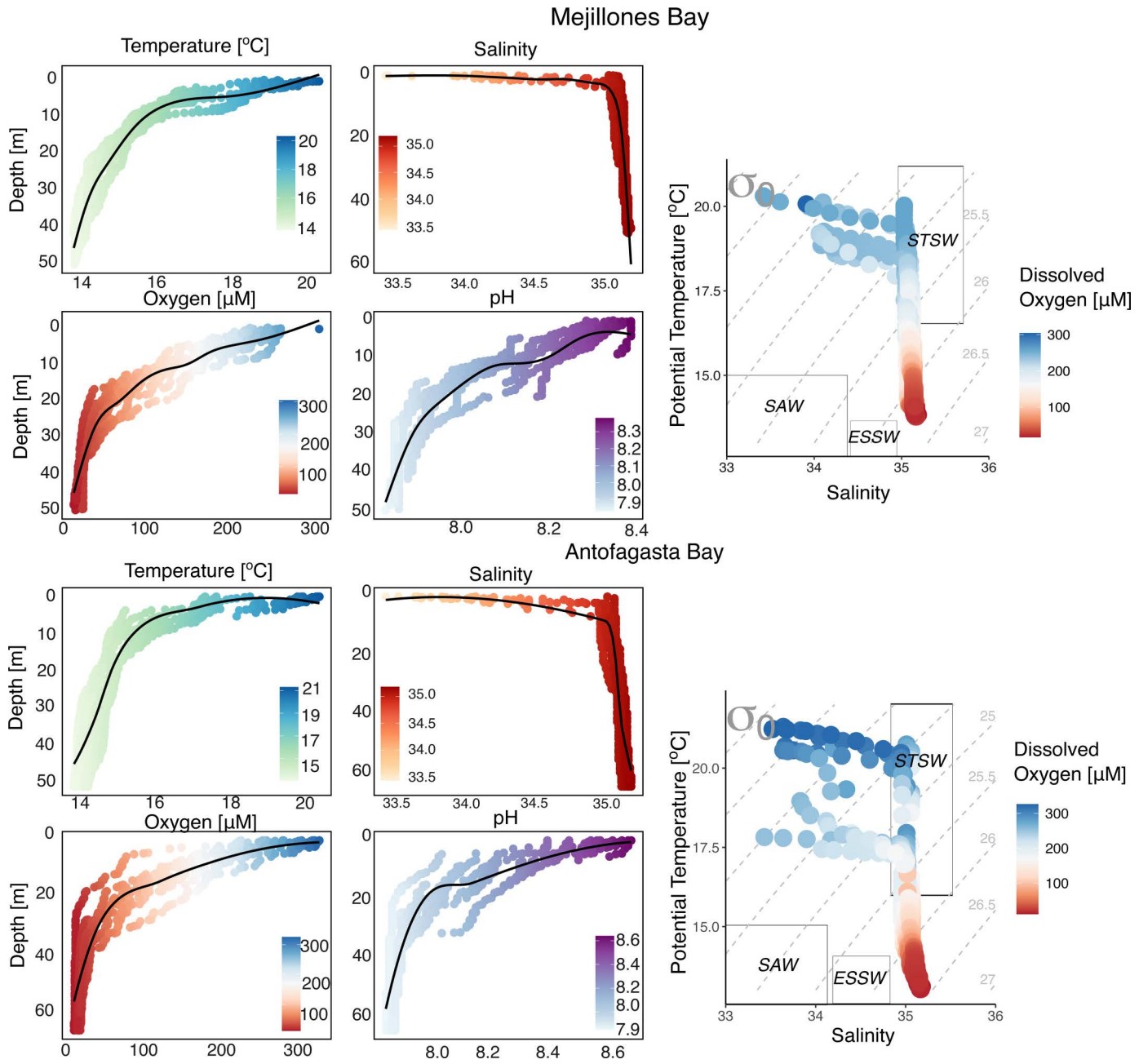

**Fig 2. Vertical profiles of temperature (°C), salinity, dissolved oxygen ( µM), pH, and T-S diagrams for MB and AB.**

**Table 1. Mean±standard deviation values for δ¹⁵N-Phe, δ¹⁵N-Thr, trophic position (TP) of metazoan, and protozoan, ΣV parameters, and downward fluxes of sinking POM collected from sediment trap samples deployed at MB and AB.**

| | δ¹⁵N Phe (‰) | δ¹⁵N Thr (‰) | TP Metazoan | TP Protozoan | ΣV | POC Downward Flux (mg/m²/day) |
|---|---|---|---|---|---|---|
| Mejillones Bay | 16.2±0.1 | 8.8±0.2 | 1.9±0.6 | 2.4±0.6 | 2.7±0.2 | 9±2 |
| Antofagasta Bay | 15.1±0.1 | 11.3±0.3 | 1.7±0.4 | 1.3±0.6 | 1.9±0.2 | 11±1 |

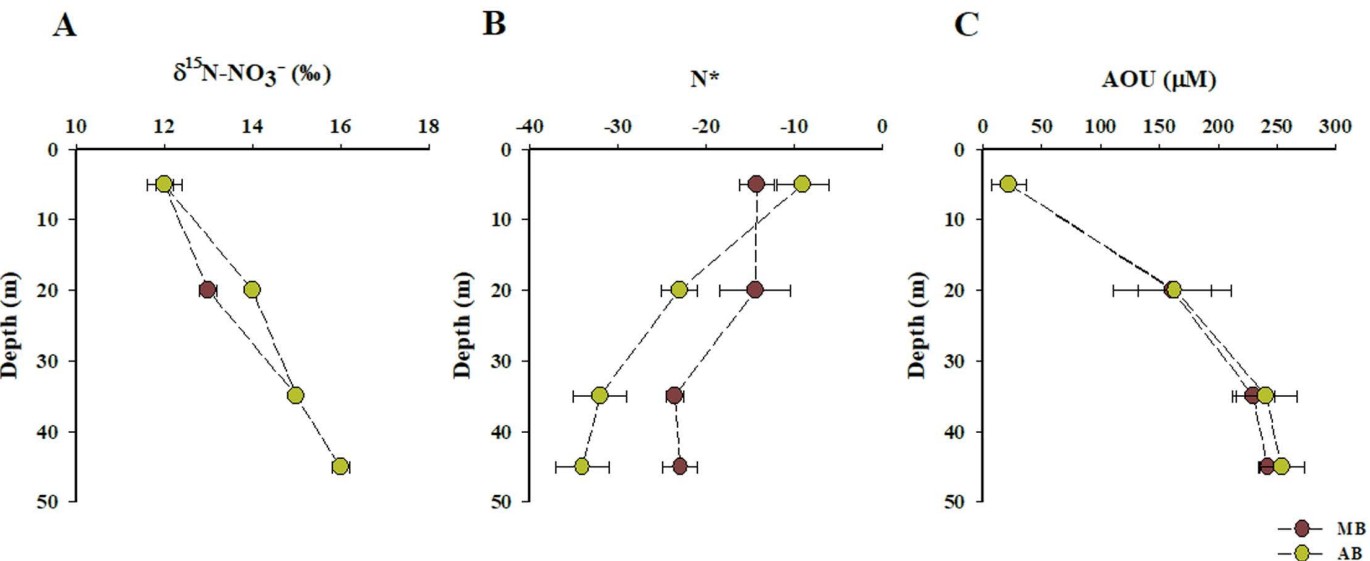

**Fig 3. Vertical profiles of (A) δ¹⁵N-NO₃⁻, (B) N*, and (C) apparent oxygen utilization (AOU) in the water columns of MB and AB.** Data are presented as mean±standard deviation values.

### 3.6. Natural abundance of δ¹⁵N-NO₃⁻ isotopes

In MB and AB, the δ¹⁵N values of dissolved NO₃⁻ ranged from 12.8±0.3 ‰ to 15.9±0.5 ‰, with a mean value of 14.8±1 ‰. No significant differences were observed between the two bays (Mann-Whitney test; $p=0.6$). Both bays exhibited a pronounced enrichment trend with depth, averaging an increase of 2.6±0.01 ‰ (Fig 3A). The lowest δ¹⁵N-NO₃⁻ values were recorded at the surface of MB (average δ¹⁵N-NO₃⁻=12±0.1 ‰ at a depth of 5 m), while the highest value was measured at 45 m depth in AB (average δ¹⁵N-NO₃⁻=15±0.01 ‰) (Fig 3A).

### 3.7. Vertical patterns of δ¹⁵N in trophic and source amino acids within suspended and sinking POM

In MB, the δ¹⁵N Tr-AA —including Ala, Val, Leu, Ile, Pro, and Glu—in suspended POM ranged from 13.4±0.3 ‰ (Leu) to 27.0±0.1 ‰ (Pro) (S2 Table in S1 Data). In contrast, the δ¹⁵N Src-AA, which includes Ser, Phe, Tyr, and Lys, varied from 8.9±0.2 ‰ (Tyr) to 18.7±0.2 ‰ (Tyr) (S2 Table in S1 Data). For sinking POM, δ¹⁵N Tr-AA values ranged from 17.9±0.2 ‰ (Leu) to 26.3±0.3 ‰ (Glu), while δ¹⁵N Src-AA values ranged from 12.9±0.3 ‰ (Tyr) to 19.3±0.5 ‰ (Lys) (S3 Table in S1 Data).

The vertical profiles of δ¹⁵N-Phe normalized Tr-AA averaged 6.1±2.7 ‰, with individual values ranging from -0.4±0.1 ‰ (Ile) to 11.9±0.4 ‰ (Glu) (S5A Fig in S1 Data). At a depth of 5 meters, the average δ¹⁵N-Phe normalized Tr-AA was 5.5±2.5 ‰, which was comparable to the mean value observed at 45 meters (5.1±2.9 ‰; Wilcoxon test; $p=0.145$) (S5A Fig in S1 Data). Significantly higher mean δ¹⁵N-Phe normalized Tr-AA acids values were found between

depths of 20 and 35 meters, with average values of 6.9±3.1 ‰ and 7.1±5.1 ‰, respectively (Wilcoxon test; $p=0.007$ and $p<0.0001$) (S5A Fig in S1 Data). The mean values for δ¹⁵N Src-AA ranged from 12.7±2.2 ‰ to 15.5±1.7 ‰, with the highest values recorded between depths of 20 and 35 meters (S5B Fig in S1 Data).

In AB, the δ¹⁵N Tr-AA of suspended POM ranged from 12.1±0.3 ‰ (Leu) to 23.9±0.3 ‰ (Ala) (S2 Table in S1 Data). The δ¹⁵N Src-AA values spanned from 9.1±0.1 ‰ (Ser) to 17.4±0.1 ‰ (Tyr) (S2 Table in S1 Data). For sinking POM, δ¹⁵N Tr-AA ranged between 16.1±0.2 ‰ (Leu) and 22.6±0.8 ‰ (Ala), while δ¹⁵N Sr-AA values varied from 13.4 ± 0.8 ‰ (Ser) to 16.5±0.8 ‰ (Lys) (S3 Table in S1 Data). Here, δ¹⁵N-Phe normalized Tr-AA varied between -1.5±0.5 ‰ (Leu) and 13.1±0.3 ‰ (Glu). Mean δ¹⁵N-Phe normalized Tr-AA values were highest at depths of 20–35 meters, measuring 7.0±3.3 ‰ and 6.8±2.6 ‰, respectively. (S5A Fig in S1 Data). In this bay, Src-AA averaged 14.1±2.2 ‰, exhibiting a less distinct vertical pattern (S5B Fig in S1 Data).

### 3.8. Trophic position in suspended and sinking POM

In MB, TP $_{Metazoan}$ in suspended POM ranged from 1.5±0.6 to 1.9±0.5, with an overall mean value of 1.7±0.1. The vertical profiles indicated an increase in TP $_{Metazoan}$ values at depths of 20 and 35 meters, reaching 1.7±0.5 and 1.9±0.5,

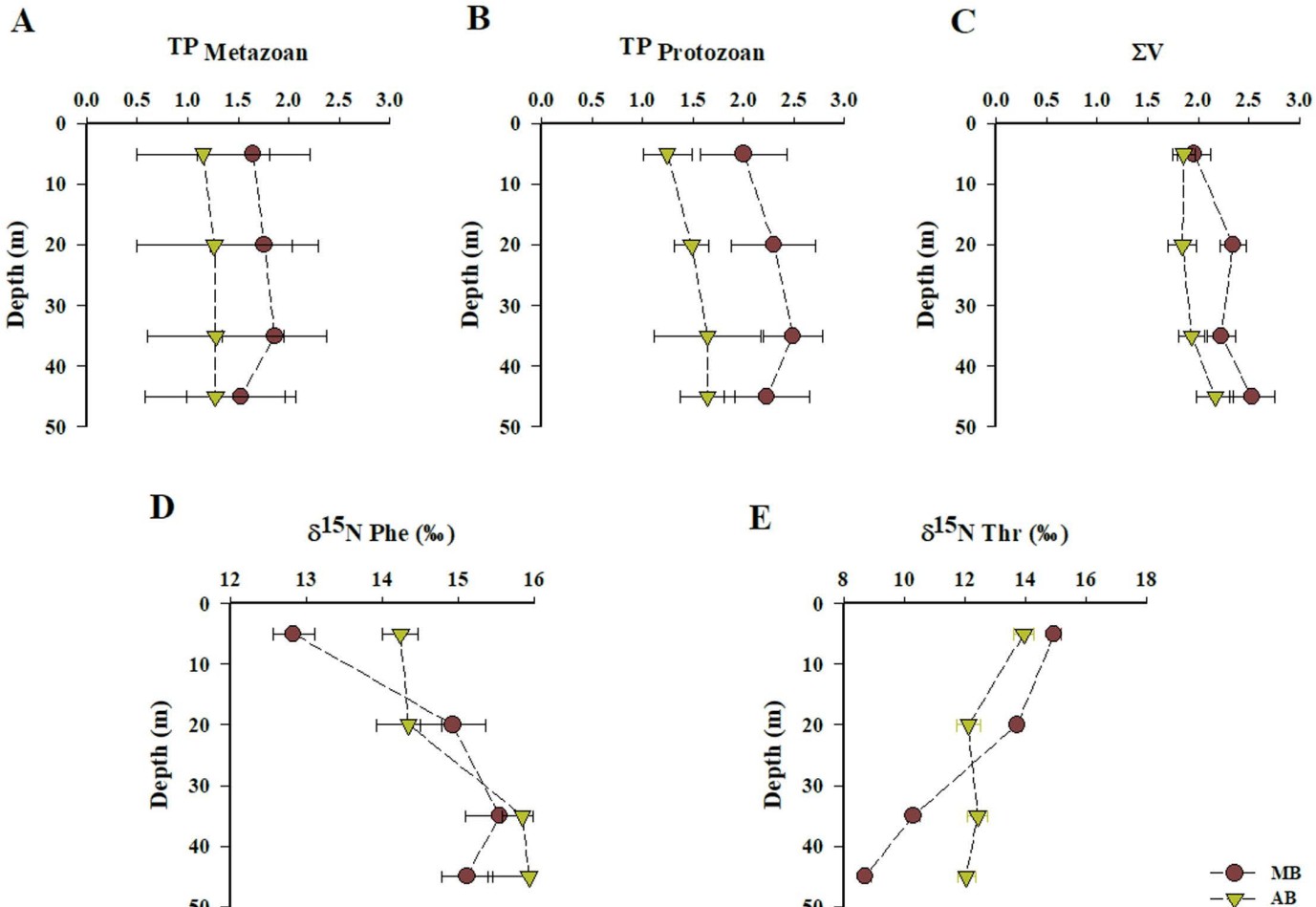

**Fig 4. Vertical profiles of (A) TP $_{Metazoan}$; (B) TP $_{Protozoan}$, (C) ΣV; (D) δ¹⁵N Phe, and (E) δ¹⁵N Thr in the water columns of MB and AB.** Data are presented as mean±standard deviation values.

respectively. However, these values declined sharply to 1.5±0.5 at a depth of 45 meters (Fig 4A). In contrast, the TP $_{Protozoan}$ ranged from 1.9±0.8 to 2.6±0.1, significantly higher than those for metazoans (Wilcoxon test; $p = 0.002$) (Fig 4B). Like the TP $_{Metazoan}$ profile, the highest values for protozoans were observed between depths of 20 and 35 meters, followed by a decrease of 0.2 ‰ at greater depths (Fig 4B). Negligible spatial differences were noted between stations for both TP $_{Metazoan}$ (Wilcoxon test; $p = 0.2$) and TP $_{Protozoan}$ (Wilcoxon test; $p = 0.3$). Sinking POM exhibited a TP $_{Metazoan}$ value of 1.9±0.8 and a TP $_{Protozoan}$ value of 2.4±0.2 (Table 1).

In AB, the estimated trophic positions for metazoans and protozoans were significantly lower than those observed in MB, with values ranging from 1.0±0.6 to 1.3±0.6 for metazoans and from 1.0±0.8 to 1.6±0.1 for protozoans (Mann-Whitney test; $p = 0.00001$) (Figs 4 A and B). The vertical profile trends in AB were less pronounced compared to those in MB. Additionally, no significant spatial differences were detected within AB (Wilcoxon Test; $p = 0.3$). For sinking POM, a TP $_{Metazoan}$ value of 1.3±0.6 was recorded, while the TP $_{Protozoan}$ value was 1.9±0.2 (Table 1).

### 3.9. Heterotrophic resynthesis in suspended and sinking POM

In MB, the average $\Sigma V$ value for suspended POM was 2.3±0.1, significantly higher than the 1.9±0.3 recorded in AB (Mann-Whitney test; $p = 0.013$). Negligible differences were observed among the sampled stations (Wilcoxon test; $p = 0.3$), with values ranging from 1.5±0.2 to 2.7±0.1. The highest average $\Sigma V$ values were recorded at depths of 20 m (2.3±0.1) and 45 m (2.5±0.2) (Fig 4C). For sinking POM, a $\Sigma V$ value of 2.7±0.2 was recorded (Table 1).

In AB, the $\Sigma V$ values in suspended POM ranged from 1.4±0.2 to 2.4±0.2, indicating negligible spatial differences within the bay (Wilcoxon test; $p = 0.2$). In contrast to MB, the highest mean values in AB were observed at greater depths of 35 m (1.9±0.2) and 45 m (2.2±0.2) (Fig 4C). Meanwhile, sinking POM exhibited an average $\Sigma V$ value of 1.9±0.2 (Table 1).

### 3.10. Spatial distribution and vertical profiles of δ¹⁵N in phenylalanine and threonine

In the MB area, the δ¹⁵N-Phe values of suspended POM demonstrated remarkable consistency, suggesting minimal spatial variations (Wilcoxon test; $p = 0.3$). These values ranged from 12.5±0.4 ‰ to 17.1±0.1 ‰ (Fig 4D). The mean values showed a gradual increase from a depth of 5 m to 35 m, peaking at 15.5±0.4 ‰, followed by a slight decline to 15.1±0.1 ‰ at a depth of 45 m (Fig 4D). In contrast, the sinking POM exhibited a δ¹⁵N-Phe value of 16.2±0.1 ‰ (Table 1). In AB, the average δ¹⁵N-Phe values in the upper 20 meters were 14.3±0.2 ‰, followed by a notable increase with depth, reaching 15.9±0.5 ‰ at 45 meters (Fig 4D). In this bay, sinking POM recorded a δ¹⁵N-Phe value of 15.1±0.1 ‰ (Table 1).

Contrasting vertical patterns of ¹⁵N fractionation in Thr were identified between Mejillones and Antofagasta bays (Fig 4E). In MB, δ¹⁵N values for Thr demonstrated a significant decrease with depth, exhibiting a net depletion of 6.2±0.2 ‰ from 5 meters to 45 meters. Specifically, δ¹⁵N-Thr values ranged from 14.9±0.2 ‰ at the surface to 8.7±0.2 ‰ at 45 meters (Fig 4E). In contrast, AB displayed a more pronounced depletion in the 5-to-20-meter depth range, with a decrease of 1.8±0.2 ‰. Below 20 meters, δ¹⁵N-Thr values stabilized, averaging 12.2±0.3 ‰ (Fig 4E).

## 4. Discussion

### 4.1. Dynamics of δ¹⁵N-NO₃⁻ in Mejillones and Antofagasta bays

The elevated δ¹⁵N-NO₃⁻ values in the waters of Mejillones and Antofagasta bays exceed the average δ¹⁵N-NO₃⁻ observed in Pacific oceanic waters (approximately 5‰) but align with values reported in coastal regions impacted by an oxygen minimum zone (OMZ) off northern and central southern Chile, as well as in OMZ waters of the Eastern Tropical North Pacific [70–74].

The observed lowest surface δ¹⁵N-NO₃⁻ levels (Fig 3A) may stem from the nitrification of NH₄⁺ excreted by zooplankton or the remineralization of fresh phytoplankton material. These processes could obscure the anticipated increase in δ¹⁵N associated solely with algal uptake [75]. Both bays showed fully oxygenated surface layers, marked by a minimum NO₃⁻ zone and chlorophyll-*a* maxima indicative of upwelling shadow and trap bays (Figs 2 C and H; S1 figs A and B; S2

figs C and D in S1 Data). This observation suggests that the photosynthetic assimilation of $NO_3^-$ plays a significant role in shaping the $\delta^{15}N\text{-}NO_3^-$ patterns in surface oxic waters. Consequently, surface $\delta^{15}N\text{-}NO_3^-$ is influenced by both the $\delta^{15}N$ of upwelled $NO_3^-$ and the subsequent absorption and assimilation of $NO_3^-$. In the deeper layer (20–45 m depth), the fractionation of N isotopes appears to be associated with denitrification, leading to a significant enrichment of $^{15}N$ in the residual $NO_3^-$ that is subsequently upwelled to the surface [18,70,71,76].

Physical and hydrodynamic forces may influence the vertical distribution of $\delta^{15}N\text{-}NO_3^-$ in both bays. The observed hydrodynamic interactions between Subtropical Surface Water (STSW) and Equatorial Subsurface Water (ESSW) (Fig 2) may have diluted the surface $\delta^{15}N\text{-}NO_3^-$ signal, resulting in the homogenization of $\delta^{15}N$ in upwelled $NO_3^-$ as deeper waters migrate from the core of the OMZ to more oxygenated surface waters [73].

### 4.2. Isotopic evidence for marine-derived, locally sourced POM

The $\delta^{15}N\text{-}Phe$ values measured within the upper 20 meters of the water column in both bays (mean value: 14.1 ± 0.4 ‰) were comparable to the mean surface $\delta^{15}N\text{-}NO_3^-$ values observed (12.4 ± 0.3 ‰ and 13.3 ± 0.4 ‰) (Figs 3A and 4D). This similarity suggests that the POM is derived from local photosynthetic sources, primarily sustained by upwelled $NO_3^-$ enriched in $^{15}N$ from deeper, oxygen-depleted waters. Consequently, we infer that *in situ*-formed POM predominates in the POM pool of the study area. In these waters, primary productivity is largely confined to the upper 20 meters of the water column [40], which aligns both conceptually and operationally with the observed vertical distribution patterns of chlorophyll-*a* and the $\delta^{15}N$ isotopic values of Phe and $NO_3^-$ in both bays (Figs 3A and 4D; S1 Figs A and B in S1 Data).

The potential contributions of organic matter of anthropogenic origin cannot be ruled out given the intense anthropogenic pressure to which both bays are subjected, which is related to mining industry port operations, desalination plants, artisanal fisheries, artificial touristic beaches, and sewage discharge [77–79]. The mole percent weighted sum of $\delta^{15}N$-THAA values of POM in the study area, considered a proxy for bulk $\delta^{15}N$ POM value (see Methods section for the operational definition), were 18.5 ± 1.8 ‰ in MB and 16.9 ± 1.3 ‰ in AB (S6A Fig in S1 Data).

These elevated $\delta^{15}N$ values in the suspended and sinking POM are expected for waters influenced by OMZ [55,72,73], and fall within the range of $\delta^{15}N$ of particulate N observed in OMZ waters off central Chile [80]. Additionally, they are similar to values observed in wastes that have experienced anthropogenic disturbances, which are isotopically rich in heavy N components, such as those derived from human wastewater and livestock (e.g., $\delta^{15}N$ of 10‰ to 22‰) [81,82], which makes it difficult to discriminate between anthropogenic and natural sources using only our proxy-bulk $\delta^{15}N$ -THAA data. However, our auxiliary mole percent weighted sum $\delta^{13}C$-THAA data were less ambiguous, yielding values of -21.7 ± 1.3 ‰ and -20.2 ± 1.1 ‰ for MB and AB, respectively (S6B Fig in S1 Data), which are $\delta^{13}C$ values commonly assigned to organic matter of marine origin in mid- and low-latitude regions [83–85], and align with the $\delta^{13}C$ bulk POC values observed in coastal upwelling and epipelagic waters of Chile [86,87]. This evidence reinforces our inferences drawn from the $\delta^{15}N\text{-}NO_3^-$ and $\delta^{15}N\text{-}Phe$ data regarding the origin and sources of POM in the study area, confirming the dominance of marine-derived POM in the waters of both bays. The similar $\delta^{15}N$ enrichment patterns noted in $NO_3^-$ and Phe suggest that this POM is primarily sourced from local origins, dependent on regenerated N from upwelling processes. The importance of N regeneration in upwelling regions has been previously documented, underscoring its role in supporting marine productivity [88–91].

### 4.3. Major sources of suspended and sinking POM

Given the diverse potential sources of organic matter in the surface waters of MB and AB, the multi-source percentage contribution model of Stock et al. [92] was applied to our $\delta^{15}N$ isotope dataset. In MB, at depths of 5–20 m, phytoplankton (~48%) and MDOM (~39%) were the primary contributors to POM signature, followed by zooplankton and fecal pellets (Fig 5A). At 35–45 m, the isotopic contributions from the sources were more evenly distributed (Fig 5B). In AB, the general pattern remains similar to that observed in MB. However, at depths of 5–20 m, our results reveal a more pronounced

dominance of phytoplankton (~62%) compared to other sources (Fig 5C). This increased contribution of phytoplankton persists at depths of 35–45 m (Fig 5D).

In MB, the isotopic signal of phytoplankton dominates the surface layers, but their contribution decreases at greater depths, likely due to grazing and bacterial heterotrophic consumption, as suggested by the second significant contribution from MDOM (Fig 5A). This shift allows other sources to contribute more significantly to the organic matter pool in subsurface layers (Fig 5B). Therefore, although previous studies in MB have suggested a significant degree of coupling between heterotrophic prokaryote production and gross primary production [93], as well as the importance of microbial grazing in carbon export [94], our isotopic results suggest that this relationship may be more dynamic than previously expected. In contrast, AB exhibits a consistent pattern across both layers (Figs 5 C and D), likely driven by a combination of faster-sinking particles and/or less efficient remineralization compared to MB, as has been previously reported for the region

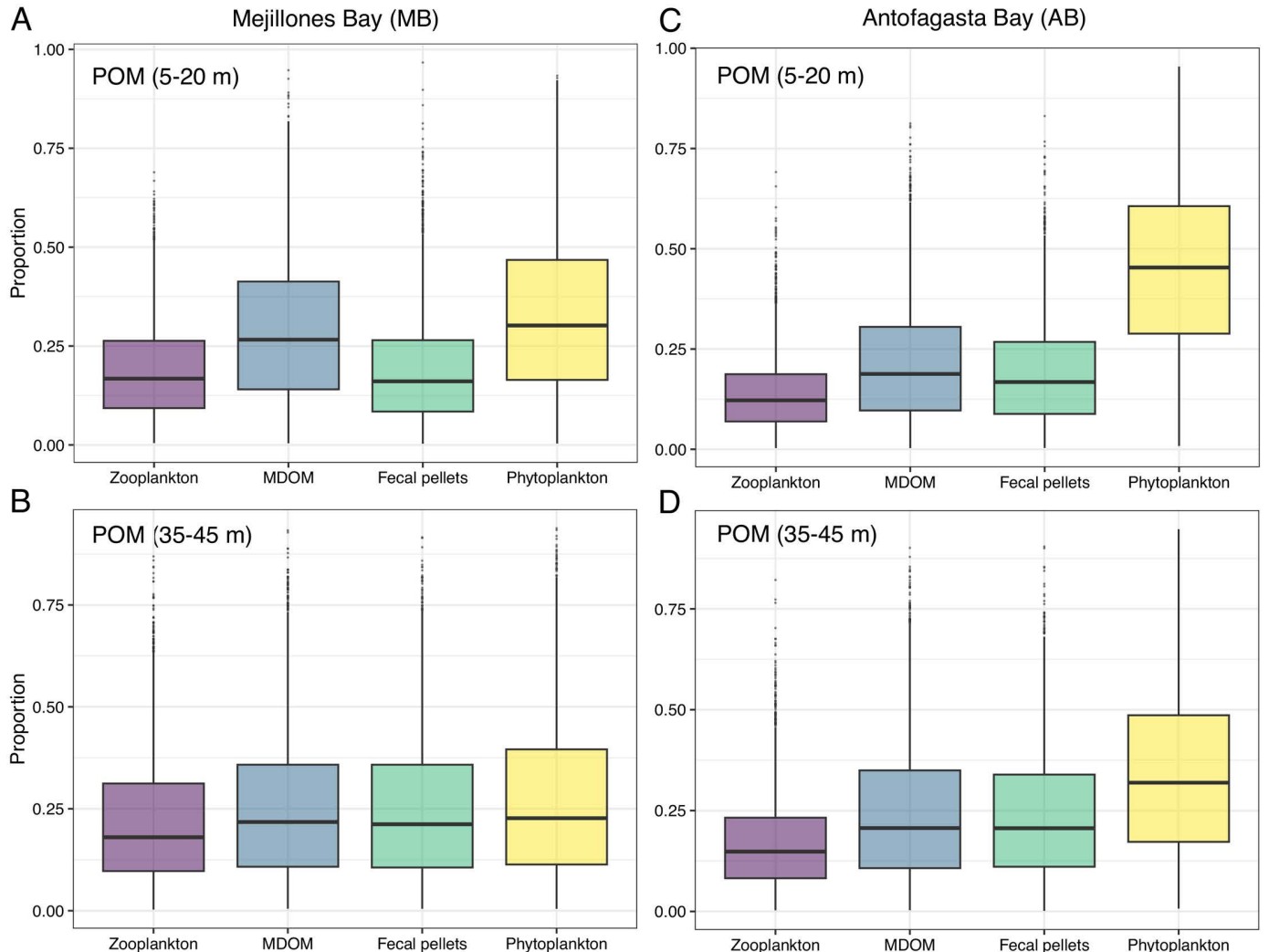

**Fig 5. Relative fractional contribution of POM from the main end-members (i.e., zooplankton, MDOM, fecal pellets, and phytoplankton) based on $\delta^{15}N$ at varying depths (5–20 m and 35–45 m) in MB and AB, estimated with the MixSIAR mixing model (Stable Isotope Analysis in R).**

[95,96]. Overall, these results suggest that surface suspended POM in both bays can reflects local primary productivity and upwelling dynamics, which shape its composition and subsequent transformation in subsurface layers.

Using the MixSIAR model with a different configuration, we assessed the isotopic contribution of all end-members to 45 m sediment traps in both bays, including the reported water column POM. The results indicate a fairly uniform $\delta^{15}N$ signal, with no end-member contributing more than ~25%, suggesting that exported material reflects a mix of surface processes (Figs 6 A and B). We acknowledge that these differences may change with the relaxation and activation of upwelling, and/or that the productivity cycles of these bays could be decoupled, potentially altering their contribution to the isotopic signal. Nevertheless, our findings offer valuable insights into the processes influencing surface POM and its contribution to subsurface layers (see more details of the MixSIAR analysis in S7 Fig in S1 Data). Future studies targeting the molecular composition and degradation state of suspended and sinking POM could provide further insights into its ecological and biogeochemical roles in these dynamic coastal environments.

## 4.4. Vertical patterns of trophic position and resynthesis of POM

The $\delta^{15}N$-amino acid data indicate that trophic position and $\Sigma V$ parameter values closely align with the typical vertical distribution of photosynthetic and microbial activity in these waters, modulated by the vertical physical and chemical gradients characteristic of the upwelling ecosystem in Chilean coastal waters [80,90,97–104]. In the water columns of both bays, the TP $_{Metazoan}$ values fell within the expected ranges for primary producers (1.0) and primary consumers (2.0) [105] (Fig 4A) indicating limited trophic transfer of POM in metazoans. These values align with prior observations of low and high molecular weight dissolved organic nitrogen in the North Pacific Subtropical Gyre [105], as well as with trophic position values for POM reported in the Bermuda Atlantic Time-Series Study [106]. Additionally, they align with findings by Vargas and González [94], who noted that the impact of zooplankton grazing on primary producers was relatively low in northern Chilean coastal waters. In contrast, TP $_{Protozoan}$ values were significantly higher than those for metazoans in MB and AB (Wilcoxon test; $p = 0.002$ and $0.004$, respectively) (Figs 4 A and B), providing direct evidence for the central role of protozoan heterotrophy in these upwelling bays. In MB, TP $_{Protozoan}$ averaged $2.3 \pm 0.3‰$, while in Antofagasta Bay,

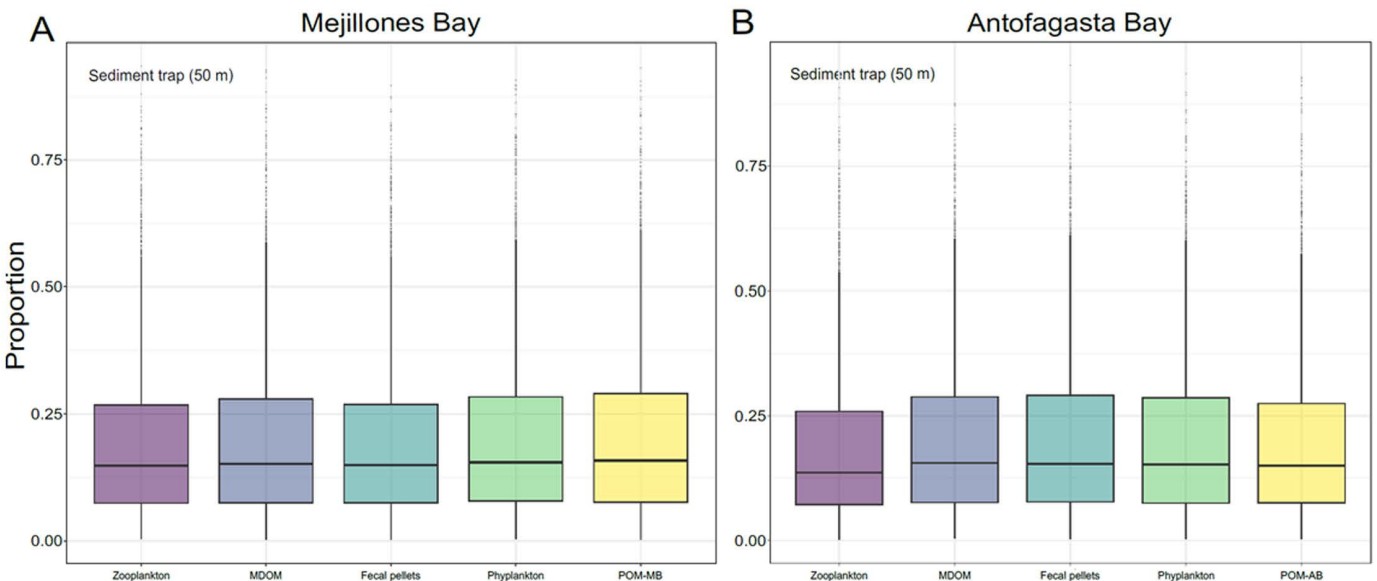

**Fig 6. Relative fractional contribution of POM to sediment traps from the main end-members (zooplankton, MDOM, fecal pellets, phytoplankton, and water column POM), based on $\delta^{15}N$ values.** Results for (A) MB and (B) AB were estimated by the MixSIAR mixing model.

the average value was significantly lower at 1.5±0.3‰ (Mann-Whitney test; $p=0.00001$). Both bays exhibited the highest values of TP $_{Protozoan}$ and $^{15}$N-enrichments in Ala (e.g., 2.1±0.1‰) at depths ranging from 20 to 35 meters, which correspond to the chlorophyll-$a$ minimum, the highest AOU values—an indirect indicator of respiration rate magnitude [107]—, alongside hypoxic to suboxic conditions (Figs 2 G and H; Fig 3C; S1A and b Figs; S2 table in S1 Data). This suggests that a significant fraction of heterotrophic protozoan cycling of POM occurs in the oxygen-deficient water layers of the bay [60]. This finding is consistent with the observations made by Parris et al. [108], who noted that protists form the majority of the eukaryotic community within the oxycline and oxygen minimum zone (OMZ) of the Eastern Tropical South Pacific off northern Chile. In the OMZ off central-southern Chile, protozoans, including parasitic dinoflagellates, emerged as the dominant organisms. Concurrently, dysoxic and suboxic conditions promoted the enrichment of fungi and phagotrophic protists [109]. This pattern has also been observed at several other OMZ waters, including the Black Sea, Mariager Fjord, the North Pacific, Saanich Inlet, and the Gulf of Mexico [110–114]. Furthermore, studies have shown that in the coastal waters of northern Chile, protozoans remove a significant portion of primary production, surpassing the impact of zooplankton and playing a vital role in transferring organic carbon to higher trophic levels [94].

In addition, the vertical distribution and $\Sigma V$ values indicate extensive bacterial reworking of POM throughout the water column, including the photic zone. This highlights the critical role of bacterial heterotrophic resynthesis in POM cycling. $\Sigma V$ values in both bays ranged from 1.0 to 2.6, indicating substantial salvage incorporation and amino acid resynthesis [20]. The increasing of TP $_{Protozoan}$ and $\Sigma V$ values with depth, combined with decreasing dissolved oxygen levels, lower N* values, and higher $NO_2^-$ concentrations (Figs 2 C and H; Fig 3B; S2A and B Figs in S1 Data) strongly indicate that the microbial assemblage thriving in oxygen-depleted conditions are key contributors to and mediators of POM cycling in these waters. In both bays, the $\delta^{15}$N values of Tr-AA were higher bellow 20 m compared to the values observed at surface (S5A Fig; S2 table in S1 Data). These observations align with previous studies that documented a significant increase in the $^{15}$N content of suspended particles with depth in both marine and lacustrine water columns [18,27,28,34]. The enrichment of heavier N isotope is attributed to the degradation of labile organic material, which is depleted in $^{15}$N, leading to the presence of suspended particles that are enriched in $^{15}$N [34]. In the study area, estimates of trophic transfer and amino acid resynthesis strongly suggest that the enrichment of $^{15}$N in amino acids is primarily influenced by a microbial loop, particularly intense at the base of the oxycline. We infer that these processes encompass complex trophic interactions, including protozoan grazing and the microbial heterotrophic reworking and resynthesis of suspended and sinking POM within the water columns of Mejillones and Antofagasta bays. Recent studies by Srain et al. [80] reveal a complex heterotrophic microbial consortium in the suboxic and anoxic coastal waters of upwelling ecosystem off central-southern Chile, confirmed through incubation experiments and field measurements.

Notably, in both bays, the $\delta^{15}$N-Phe values in POM showed an increase with depth, particularly between 20–35 meters (i.e., the base of the oxycline) (Fig 4D). While it has been widely shown that $\delta^{15}$N-Phe values exhibit minimal fractionation in experimental cultures with fungi, bacteria, and archaea utilizing organic substrates [20,115], changes in $\delta^{15}$N-Phe values can occur when peptide bonds are broken during the microbial degradation of organic matter [35]. The synchronous $\delta^{15}$N enrichment of Phe-normalized Tr-AA and Phe with depth (Fig 4D; S5A Fig in S1 Data) between 20–35 meters depth aligns with the findings of Hannides et al. [34], suggesting a hypothetical $^{15}$N enrichment resulting from the degradation of suspended organic matter via external hydrolysis. Extracellular enzyme activity has been observed in various marine particles, including suspended particulate matter [116,117], marine snow [118,119], and sinking particles collected in sediment traps [120,121]. In the oxic and suboxic waters of northern Chile, microbial peptide hydrolysis occurs at rates comparable to or exceeding amino acid uptake in the water column [98,122].

## 4.5. Enhanced protozoan trophic transfer and bacterial resynthesis of POM in denitrifying water layers

In both bays, at depths of approximately 20–45 m, anomalies in dissolved inorganic nitrogen (Fig 3B), were likely linked to microaerophilic and anaerobic metabolism within the nitrogen cycle of the water column, coinciding with the highest

TP $_{Protozoan}$ and ΣV values (Figs 4 B and C). The N* values indicated a deficit in $NO_3^-$ across both oxyclines, mirroring the negative N* values observed in the upwelling system between 21° and 33°S, which have been linked to denitrification and anammox processes [73,102,103]. Furthermore, the observed increases in $NH_4^+$ and $PO_4^{3-}$ concentrations, along with AOU and reductions in pH and TOC with depth (Figs 2 D and I; Fig 3C; S1 C and D Figs; S3A-D Figs in S1 Data), support the hypothesis of intensified heterotrophic recycling of POM at the base of the oxycline, where the depletion of dissolved oxygen is most pronounced.

Previous studies have reported higher protozoan grazing rates on bacteria and a greater diversity of picoeukaryotes in the anoxic waters of the Baltic Sea, the suboxic waters of MB, and the upwelling system off central-southern Chile [123,124]. These findings support our estimates of the trophic positions of protozoans based on δ$^{15}$N amino acids, underscoring the important role these organisms play in transferring POM within the denitrifying water layers of northern Chilean upwelling bays. Furthermore, our results are consistent with those reported by González and Quiñones [125], who found that microbial communities in the oxygen-depleted waters of Chile exhibit similar, if not greater, heterotrophic metabolic potential compared to those in more oxygenated layers. This observation corroborates earlier studies in the coastal waters of northern Chile, which document substantial temporal and vertical variability in nitrogen, phosphorus, and silicate levels, as well as N and Si ratios, indicating dynamic recycling processes of organic matter throughout the water column [39,42].

The vertical profiles of apparent nitrogen isotopic fractionation (εx/Glu) in MB corroborate these findings. Gly and Ser exhibited significantly more negative fractionation, particularly near the base of the oxycline, with average values of -11.6 ± 1.3‰ and -11.7 ± 0.9‰, respectively, in comparison to those observed in AB (Mann-Whitney Test; $p = 0.03$) (Figs 7 A and B; S4 Table in S1 Data). These results suggest a weaker connection to Glu metabolism and the involvement of a broader range of metabolic pathways [20]. This is indicative of microbes in low-oxygen environments, where extreme conditions drive these communities toward diverse metabolic adaptations [108,126–128]. Further, the overall $ε_{x/Glu}$ patterns indicate that assimilation and re-synthesis of preformed amino acids were dominant over *de novo* synthesis in the denitrifying waters in both bays [20] (S4 Table in S1 Data).

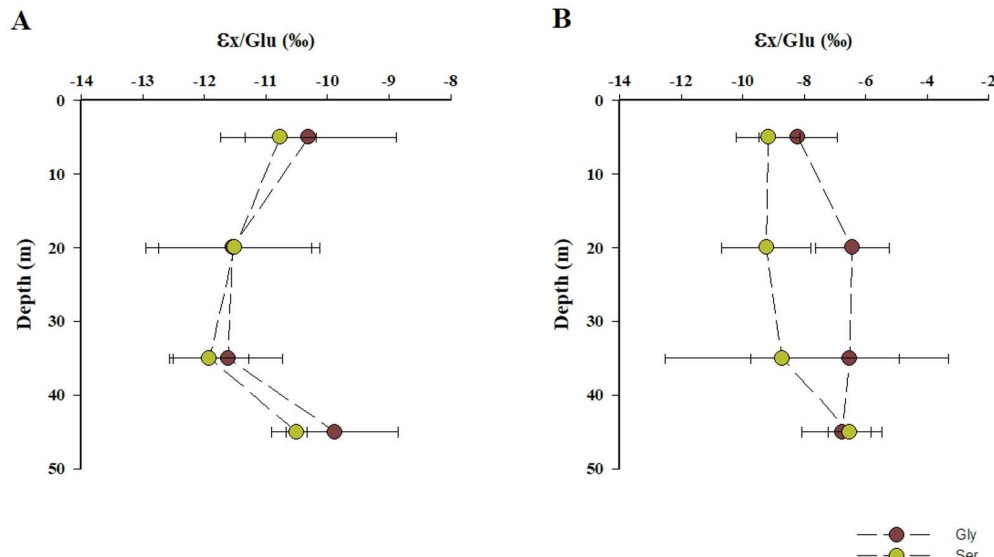

**Fig 7. Vertical patterns of apparent isotopic fractionation (εx/Glu) for Gly and Ser in the water columns of (A) MB and (B) AB.** Data are presented as mean ± standard deviation values.

The observed vertical variation in $\varepsilon_{x/Glu}$ values likely reflects the interplay of distinct metabolic pathways involved in amino acid assimilation and re-synthesis throughout the water column [20,129–131]. This variation aligns with the significant differences in microbial metabolisms between oxic and oxygen minimum zone waters, driven by varying oxygen availability [126,127,132,133].

## 4.6. Spatial variability in trophic transfer and resynthesis of POM: The role of interactions between biological and physical processes

The vertical physicochemical structure of the water column in both bays shows remarkable similarity in terms of temperature, salinity, dissolved oxygen, and pH (Figs 2 A-J; Mann-Whitney test; $p > 0.05$). Additionally, the absence of significant correlations between $TP_{Metazoan}$, $TP_{Protozoan}$, and $\Sigma V$ with temperature, salinity, dissolved oxygen, pH, and chlorophyll-$a$ contents (Spearman r; $p > 0.05$; S5 Table in S1 Data) suggests that other unexamined factors or their interactions may account for the observed spatial variations in the heterotrophic POM cycling. MB exhibited significantly higher values of $\Sigma V$ of POM compared to AB. This disparity is likely attributable to differing supplies of bioavailable amino acids between the two bays [21]. Such variations may be driven by biological processes, including primary production and grazing, as well as physical factors like water mixing and sediment resuspension, all of which influence the concentration and form of these compounds in marine environments [134,135].

In MB, the highest observed trophic position values for metazoans associated with POM, combined, with a significant decrease in $\delta^{15}$N-Thr at greater depths, may indicate intensified mechanical degradation of POM by the zooplankton community. This increased degradation could result in a greater supply of zooplankton waste material, which, in turn, may enhance protozoan grazing intensity. This intensified grazing could lead to a higher availability of amino acids compared to AB. This process involves the consumption of suspended and sinking organic matter, which releases amino acids and nutrients back into the water column, thereby facilitating and enhancing the incorporation and resynthesis of these essential compounds [136,137]. In MB, the Phe-normalized $\delta^{15}$N values for Ala and Thr in suspended POM were found to be comparable to those of isotopically defined organic matter end-members [67]. This includes MDOM in shallower water layers (5–20 meters depth; ANOSIM; R = 0.3) and both MDOM and FP) in deeper water layers (35–45 meters depth), where MDOM values surpassed those of FP (ANOSIM; R = -0.02 and 0.1, respectively). (S8 Fig; S6 Table in S1 Data).

Additionally, the sinking POM collected from traps in MB was similar to that of zooplankton, MDOM and FP, following a similitude hierarchy of FP > MDOM > Zoo (ANOSIM; R = -0.2, 0.03 and 0.2 respectively; p = 0.00001) (S5 Table in S1 Data). This suggests that zooplankton waste, along with microbial degradation and the reworking of organic matter, plays a crucial role in mediating POM cycling. This stands in stark contrast to the findings in AB, where no significant similarities were observed between POM and Phe-normalized $\delta^{15}$N values of Ala and Thr organic matter end members (S8 Fig; S6 Table in S1 Data).

Furthermore, in the water column of MB, the depletion of $^{15}$N in Thr with depth coincided with the $^{15}$N enrichment of Phe and the highest trophic positions and $\Sigma V$ values. These N isotopic fractionation patterns suggest an enhanced microbial solubilization of particles through enzymatic extracellular hydrolysis, particularly via the cleavage of peptide bonds [34,35,138]. This process may enrich the labile nitrogen pool by enhancing heterotrophic activity and facilitating remineralization, ultimately resulting in the loss of particles [139]. This could help explain the lower rates of POM export observed in MB compared to those recorded in AB.

A comparative analysis of MB and AB revealed that MB, on average, had lower surface temperatures, reduced dissolved oxygen inventories, greater water column stratification, and higher concentrations of $NO_2^-$ compared to AB (Figs 8 A-H). Distinct differences in the spatial distributions of these parameters were observed between the two bays. In MB, colder surface temperatures were recorded at the head, while the warmest temperatures were observed in the central region (Fig 8A). A similar pattern was noted for dissolved oxygen inventories, which indicated that the head of the bay was the most oxygen-depleted area (Fig 8B). Increased stratification was documented from the central area toward the mouth

of the bay (Fig 8C), with $NO_2^-$ concentrations spatially coinciding with lower oxygen inventories (Fig 8D). In contrast, AB exhibited higher surface temperatures at its head (Fig 8E), and the spatial distribution of dissolved oxygen was less distinct, featuring a notable hotspot in the central region (Fig 8F). Furthermore, the water column in AB displayed lower levels of stratification (Fig 8G).

In MB, the significantly greater degree of water column stratification may enhance particle retention by slowing their descent to the seafloor. This reduced mixing creates low-turbulence zones that allow organic particles to remain suspended for extended periods, providing prolonged exposure to heterotrophic microbes that dissolve and release POM into the dissolved organic matter fraction [139–141]. Furthermore, increased stratification likely hindered the entry or exchange of more recalcitrant amino acids bound to silicate or carbonate biominerals through resuspension These amino acids may represent a significant portion of the total hydrolysable amino acids found in marine sediments [142].

Supporting evidence for this hypothesis comes from studies in the water column of AB, where sorption interactions between organic matter and minerals have been shown to provide steric protection, effectively limiting enzymatic access [96]. Therefore, increase in recalcitrant organic matter can lead to lower rates of heterotrophic metabolism due to the difficulty microorganisms face in breaking down these complex molecules, resulting in slower energy and nutrient release [143–145].

In the stratified waters of MB, elevated levels of $NO_2^-$ indicate a more intense heterotrophic processing of organic matter via denitrification (Mann-Whitney test; $p = 0.04$) (Figs 8D and D). In these waters, the differences in density between layers may create an effective barrier that prevents the mixing of oxygen-rich surface waters with deeper layers. This separation not only limits oxygen diffusion but also extends the duration of nutrient and organic compounds exposure for microbes, potentially promoting a high intensity of the denitrification process [135,146,147].

These variable conditions may have influenced the observed spatial variations in the isotopic fractionation of [15]N amino acids in these waters. Therefore, it is essential to recognize that even minor changes in the intensity of water column stratification can significantly impact the cycling of POM. Consequently, we hypothesize that the observed spatial variation in POM trophic transfer, reworking, and resynthesis arises from a complex interplay of biological and physical processes.

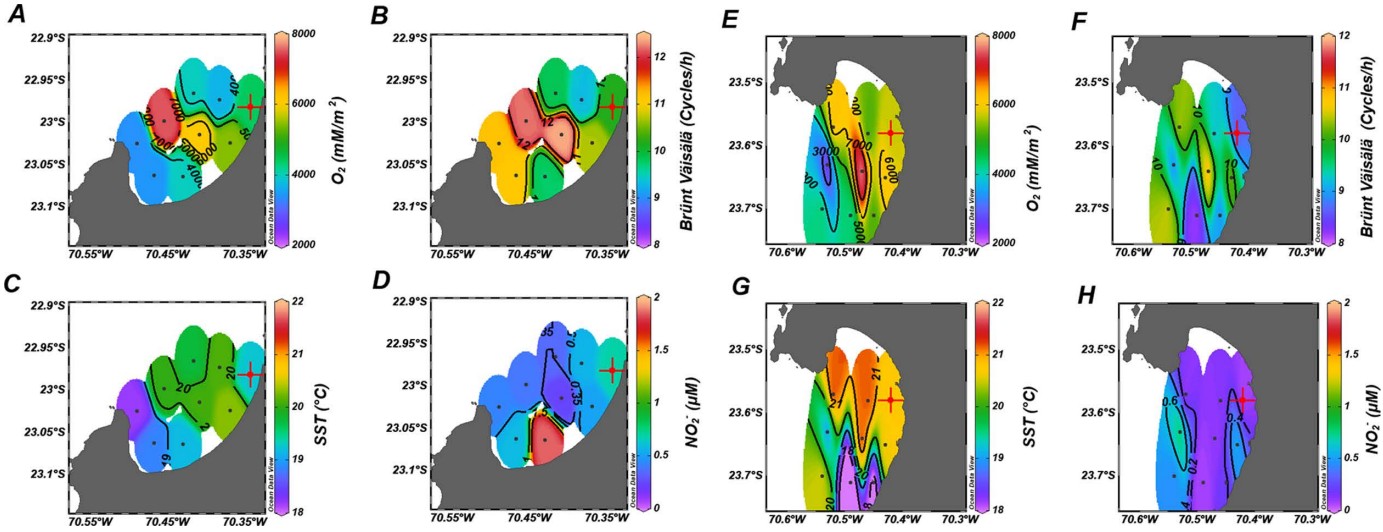

**Fig 8. Spatial distributions of mean oxygen inventories, stratification values, sea surface temperatures (SST), and average $NO_2^-$ concentrations are presented for MB (A-D) and AB (E-H).**

Previous studies have demonstrated that the orientation, geometry, size, and geomorphology of coastal formations affect physical and hydrodynamic factors, which in turn modulate biological processes like primary productivity, community respiration, and the retention of planktonic propagules [10,106,148,149]. This study demonstrates that geographic and topographic variables can generate contrasting oceanographic and hydrodynamic patterns on a local scale, significantly impacting the intensity of heterotrophic reworking and the trophic dynamics of both suspended and sinking POM. Our results challenge the paradigm that the vertical physical-chemical structure of the water column alone ultimately modulate the spatial patterns and intensities of organic carbon cycling in the coastal pelagic environments, and highlight the inter-actively influence of physical, chemical, and biological dynamics alongside with the heterogeneous nature of topographic morphology and geographic orientation of the upwelling bays systems. However, these observations warrant further local hydrodynamic and biogeochemical studies to fully understand these complex interactions.

## 5. Conclusions

This study reveals significant spatial variability in POM cycling within the contrasting coastal bays of the Humboldt Current Upwelling System. MB, characterized by stronger water column stratification and a northward geographic orientation, demonstrates enhanced heterotrophic reworking of POM. This is evidenced by significantly higher metazoan and protozoan trophic positions, increased $\Sigma V$, and a notable depletion of $\delta^{15}N$ in Thr with increasing depth. These observations align with elevated $\delta^{15}N\text{-}NO_3^-$ and $NO_2^-$ content values noted at deeper layers in MB, suggesting active denitrification processes. In contrast, AB, characterized by a less stratified water column and southward orientation, relies more heavily on surface phytoplankton for the POM pool and exhibits reduced heterotrophic processing, as indicated by lower trophic positions and $\Sigma V$ values. MixSIAR modeling consistently identifies phytoplankton as a primary source of POM in both bays; however, the subsequent processing pathways diverge sharply.

This divergence underscores the intricate interplay between biological factors (such as grazing and microbial activity) and physical factors (including stratification and water column mixing). These physical factors are influenced by the unique topographic and geographic configurations of each bay, resulting in distinct local hydrographic regimes that ultimately affect POM cycling trajectories. Our findings emphasize the necessity of considering both biological and physical influences when evaluating POM cycling in coastal upwelling systems, as generalizations based solely on hydrographic features may neglect essential biological controls. Future research should focus on identifying the specific microbial communities and processes responsible for the observed spatial differences in heterotrophic reworking. Additionally, incorporating temporal variability will deepen our understanding of POM dynamics in these highly productive and variable systems.

## Supporting information

**S1 Data. S1 Fig.** Vertical profiles of Chlorophyl-a, and $NH_4^+$ contents in MB (A and C) and AB (B and D). **S2 Fig.** Vertical profiles of $NO_2^-$, and $NO_3^-$ contents in MB (A and C) and AB (B and D). **S3 Fig.** Vertical profiles of $PO_4^{3-}$, and TOC contents in MB (A and C) and AB (B and D). **S4 Fig.** (A) A box plot accompanied by jitter points illustrates the dissolved oxygen inventories in the water columns of MB and AB. Additionally, vertical profiles of water column stratification, represented by the Brünt-Väisälä frequency, are presented for both MB (A) and AB (B). **S5 Fig.** Vertical profiles of mean values for (A) $\delta^{15}N$ Phe-normalized Tr-AA, and (B) Sr-AA in the water columns of MB and BA. Data are presented as mean ± standard deviation values. **S6 Fig.** Box and jitter plots of (A) $\delta^{15}N$ and (B) $\delta^{13}C$ of THAA in suspended POM from MB and AB. **S7 Fig.** Bayesian stable isotope mixing model (MixSIAR) applied to $\delta^{15}N$ end-members in POM samples from two depths in Mejillones and Antofagasta Bays. The top panel shows results from Mejillones Bay at two depths: 5–20 m (A-B) and 35–45 m (C-D). Panels A and C display the range of variability for the end-members, along with the POM data at their respective depths, and density plots illustrating the prior and posterior distributions for each source. Panels B and D depict the relationships (distributions and correlations) between the different sources within the mixture. The bottom panel shows the same distributions for Antofagasta Bay. **S8 Fig.** Biplot data on phenylalanine (Phe)-normalized threonine (Thr) and

alanine (Ala) $\delta^{15}N$ values for end-members (microbially degraded organic matter, MDOM; fecal pellets, FP; phytoplankton; and zooplankton) as well as suspended and sinking POM from MB and AB are presented as means ± standard deviations. Suspended and sinking POM data are from this study. Other data are from published studies, detailed in Table S1. **S1 Table.** Phe-normalized $\delta^{15}N$ values of Thr and Ala used in S7 Fig., with metadata. **S2 Table.** Full data on $\delta^{15}N$ amino acids in suspended POM collected from MB and AB. **S3 Table.** Full data on $\delta^{15}N$ amino acids in sinking POM collected from sediment traps. **S4 Table.** Full data on apparent isotopic fractionation ($\varepsilon x/Glu$) values from suspended POM collected from MB and AB. **S5 Table.** Spearman rank correlation p-values between physicochemical and $\delta^{15}N$ amino acid-derived parameters obtained from suspended POM collected from the water columns of MB and AB: $p \geq 0.1$ indicates a very weak or no correlation between variables. **S6 Table.** Analysis of Similarity (ANOSIM) between $\delta^{15}N$ Phe normalized $\delta^{15}N$ Ala and $\delta^{15}N$ Thr organic matter end members values and $\delta^{15}N$ Phe normalized $\delta^{15}N$ Ala and $\delta^{15}N$ Thr values found in suspended and sinking POM collected from MB and AB. An R value close to "1" suggests dissimilarity between groups while an R value close to "0" suggests an even distribution of high and low ranks within and between groups.
(PDF)

## Acknowledgments

We acknowledge the support provided by Professors Juan Ávila Donoso and Pedro Cortés Peña of the Chemistry, Bioinorganic, and Analytical Laboratory of the Faculty of Basic Sciences at the University of Antofagasta. We are grateful to the crew of the L/C ANAGO for their assistance during sampling, and to Marine Ecologist Maritza Malebrán for her technical support. We also extend our thanks to the anonymous reviewers for their invaluable contributions to improving this manuscript.

## Author contributions

**Conceptualization:** Benjamin Srain, Jorge Valdés, Andrés Camaño.

**Data curation:** Benjamin Srain, Edgart Flores.

**Formal analysis:** Benjamin Srain, Edgart Flores.

**Funding acquisition:** Andrés Camaño.

**Investigation:** Benjamin Srain.

**Methodology:** Benjamin Srain.

**Supervision:** Benjamin Srain.

**Validation:** Benjamin Srain, Jorge Valdés, Andrés Camaño.

**Visualization:** Benjamin Srain.

**Writing – original draft:** Benjamin Srain.

**Writing – review & editing:** Benjamin Srain, Edgart Flores, Jorge Valdés.

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
