## [Decision Letter · Decision Letter 0]

23 Jul 2024

PONE-D-24-25324ORIGIN, TROPHIC TRANSFER AND RECYCLING OF PARTICULATE ORGANIC MATTER IN THE WATERS OF TWO UPWELLING BAYS OF HUMBOLDT CURRENT SYSTEM: INSIGHTS FROM COMPOUND-SPECIFIC ISOTOPIC COMPOSITIONS OF AMINO ACIDSPLOS ONE

Dear Dr. Srain,

Thank you for submitting your manuscript to PLOS ONE. After careful consideration, we feel that it has merit but does not fully meet PLOS ONE’s publication criteria as it currently stands. Therefore, we invite you to submit a revised version of the manuscript that addresses the points raised during the review process.

We look forward to receiving your revised manuscript.

Kind regards,

Vitor Hugo Rodrigues Paiva, Ph.D.

Academic Editor

PLOS ONE

Journal Requirements:

   "This research was funded by "Asociación de Industriales de Mejillones" (CR 4800)"

5. We note that your Data Availability Statement is currently as follows: All relevant data are within the manuscript and its Supporting Information files.

Reviewers' comments:

Reviewer's Responses to Questions

**Comments to the Author**

1. Is the manuscript technically sound, and do the data support the conclusions?

Reviewer #1: Yes

Reviewer #2: Yes

2. Has the statistical analysis been performed appropriately and rigorously? 

Reviewer #1: Yes

Reviewer #2: Yes

3. Have the authors made all data underlying the findings in their manuscript fully available?

Reviewer #1: Yes

Reviewer #2: Yes

4. Is the manuscript presented in an intelligible fashion and written in standard English?

Reviewer #1: Yes

Reviewer #2: Yes

5. Review Comments to the Author

Reviewer #1: General comments:

This manuscript uses the novel compound-specific stable nitrogen isotopes of amino acids to explore the sources and cycling of suspended and sinking POM from two upwelling bays in northern Chile. The dataset represents one of the few amino acid stable isotopic data from coastal oxygen minimum zones. The combination of seawater nitrate and POM amino acid isotopic data with oceanographic data provides a better understanding of how vertical gradients of water column modulates heterotrophic processing of suspended and sinking POM. Their findings highlight the differences in trophic transfer and microbial reworking between the two upwelling bays that shared similar vertical oceanographic profiles, suggesting potential influences from differences in coastal geometry and bottom topography at local scale. Overall, this study is generally well-structured and well-written.

While the scientific approach is generally appropriate, there are several aspects in the calculations and interpretation of amino acid stable isotope data that need improvement. My main criticisms are 1) calculation of trophic transfer is outdated; 3) the grouping of source and trophic AAs are not correct; 3) Phe-normalized d15N values of Ala and Thr are not calculated; and 4) d15N-THAA is not weight-averaged. Additionally, the wording and phrasing need to be more specific, concise, and scientifically rigorous. There are instances of redundant wording, such as 'heavier d15N enrichment,' and overly long sentences. The overall structure of the manuscript could also benefit from improvement.

Specific comments:

Abstract: When summarizing the findings of d15N-NO3, d15N-Phe, ∆Tr and ∑V, consider reporting the actual numerical values instead of providing a general description that these parameters led to certain conclusions.

LN14: The authors started the abstract with a general description of pelagic POM which does not serve as a strong motivation for this study since pelagic POM has been widely studied. Instead, highlighting POM processing in the productive and oxygen-deficient upwelling systems would be more compelling for readers. Consider emphasizing the uniqueness of the study area rather than the general importance of POM.

LN16: Replace “chemical” by “characterizable”. With increasing depth, there is an increase in uncharacterizable POM in degraded organic matter, which sometimes constitutes the largest portion overall, whereas amino acids account for the largest portion in the characterizable portion.

LN21-22: The study objectives need to be more specific than simply “gaining insight”. Consider detailing objectives such as testing the differences between the two bays and/or examining the correlations between heterotrophic processing and oceanographic conditions.

LN23-24: The ranges of d15N-AA are neither important nor informative results and do not need to be reported in the abstract. I suggest reporting the average d15N of source and trophic AAs (groupings to be revised – see LN211).

LN30: Remove “at” or “through”.

LN34: Replace “this” by “the northern Chilian”

LN45: Replace “largest” by “largest characterizable”

LN54: The explanation of why d15N-AA reflects trophic transfers and degradation is not clear. The grouping of source and trophic AAs should be described here.

Methods: The overall structure of this section is disorganized. It should follow a logical order: sampling, processing, analysis, and calculations/statistics. Consider starting with the sampling of water samples and the deployment and collection of sediment traps, followed by their processing and analyses. Providing a summary paragraph detailing the parameters measured for water samples and sediment traps respectively would enhance clarity for readers.

LN111: Replace “three special stations were established” by “three stations were established for stable nitrogen isotope analysis”.

LN115: Replace “These subsamples” by “Water samples for d15N-NO3 analysis”.

LN117: Remove “Additionally, … ocean floor” and insert a paragraph/section of sediment trap deployment here.

LN143: The term “all samples” is confusing because the authors mentioned that amino acid d15N was measured only in samples from three stations. Please specify which samples from which stations were included in the analysis.

LN157: The reproducibility of C and N of amino acid stable isotopes are typically different. It would be helpful to report specific reproducibility values for C and N separately.

LN178: Standards for nitrate d15N measurements should be reported.

Sect. 2.6: This section requires major revision. Firstly, the calculation of trophic transfer appears outdated, as newer equations based on d15N of Phe and Glu (for metazoan foodwebs, Chikaraishi et al., 2009) or Ala (for protozoan foodwebs, Décima et al., 2017) and improved β values and trophic enrichment factors from meta-analyses have been reported. It is advisable to recalculate trophic positions for metazoans (using Glu/Phe) and protozoans (using Ala/Phe), especially since the authors suggest that POM transfer is predominantly facilitated by zooplankton. Secondly, the authors should also calculate Phe-normalized Ala and Thr and compare their data with previously published end-members (zooplankton, fecal pellets, degraded OM, and phytoplankton). It would be beneficial to apply a Bayesian mixing model using these end-members to estimate the composition of OM. This approach would strengthen your interpretation regarding the degree of trophic transfer and microbial reworking.

LN202: Remove the first “-”. Insert “d13C and” between “the” and “d15N”. D13C and d15N values of THAA are typically weight-averaged due to variations in the concentration of individual amino acids in POM, which affects the overall isotopic value. Authors should recalculate d13C- and d15N-THAA values based on amino acid concentration profiles. Additionally, THAA d13C and d15N may not accurately represent bulk d13C and d15N, especially for C. If feasible, authors should measure bulk stable isotopes separately and include these measurements in the study. Otherwise, all interpretations assuming that THAA represents “bulk” values should be removed from the manuscript.

LN211: Remove “Gly” and “Thr”. In more recent studies, Gly and Thr are not considered “source AAs” anymore, especially for Thr that gets depleted with trophic transfer. Most studies do not assign Gly and Thr to specific groupings.

LN224: Replace “AUO” by “AOU”

LN256: Please specify “significant differences” in which parameters and between which groups or conditions.

Results:

LN312: The authors did not interpret chlorine index in the discussion. Consider either removing this section or moving it to the supplementary materials.

Table 1: Since the authors mentioned that the reproducibility of d13C and d15N measurements is better than 1‰, rounding the d13C and d15N values to integers does not provide sufficient precision. Most studies report stable isotope values to the tenth place.

LN329-331: Given the wide ranges of d15N-AA from 5 to 27‰ and 9 to 24‰, it's impossible for the standard deviations of the average values to be 0.1‰. Due to this variability, reporting the average values may not provide meaningful information.

LN331-332: This sentence is confusing – were the differences insignificant for all the AAs between the two bays? According to Table 2, there appear to be large differences (up to 3-4‰) in certain AAs between the two bays.

LN334: Replace “most depleted” by “least enriched”.

LN335: Remove “Thr” from source AAs.

LN367: Sediment trap d15N-AA data should be included in Figure 3, with the legend clearly indicating which data are from water columns and which are from traps.

LN371: Recalculate the average after removing Gly and Thr.

LN377: Please refer to the comments for Sect 2.6.

LN399-401: This point should be discussed in the discussion section.

Table 2: Please refer to the comments for Table 1. Standard deviations should be reported alongside the means.

Discussion:

Sect 4.1: The authors spent a considerable amount of discussion speculating on how the distribution of d15N-NO3 could be affected by various processes, despite lacking direct evidence from the current study. Please consider shortening this section and placing more focus on POM.

LN430: Please specify “other marine oxygen-depleted waters”.

LN436-438: The authors discussed how upwelling of 15N-enriched nitrogen and incomplete nitrate utilization may lead to the relatively high d15N-NO3 compared to the Pacific average. However, I disagree that incomplete nitrate assimilation by phytoplankton is a major mechanism contributing to the enriched values since we do not observe a large discrepancy between surface d15N-NO3 and d15N-Phe (an indicator of phytoplankton biomass). The surface NO3 is already 15N-enriched as a result of upwelled regenerated nitrogen before being assimilated into phytoplankton.

LN453-454: The d15N-NO3 values in deep waters are highly variable between geographical locations, and comparison with the world average does not provide meaningful information. It would be more informative to compare these values with deep waters from adjacent locations or similar upwelling systems.

LN494: Start this sentence with POM values, stating that they are similar to d15N-NO3, as the focus here is on POM.

LN506-508: Please refer to comments regarding THAA and bulk d15N values for LN202. Since THAA cannot accurately represent bulk organic material and the THAA measured in this study does not constitute a full suite of AAs, direct comparisons in d15N should not be made.

LN521: The sentence starting with “Contrarily, it highlights the fact that” is confusing because it's unclear what 'it' refers to and why 'it' is considered contrary. Contrary to what?

LN527-532: This sentence is overly lengthy.

LN527: Please provide depth distributions of d15N-TrAA and Phe-normalized d15N-TrAA to support your statement regarding synchronous enrichment of d15N

LN530: External hydrolysis is a hypothesis; please be cautious with your interpretation.

LN537-539: Move the sentence to LN532 before “In oxic…”.

Sect 4.2: Please start the section with a summary sentence or paragraph. There is a lot of information in the section. Consider breaking it into two subsections: one focusing on vertical profiles and the other on spatial heterogeneity.

LN554: What are the “expected”? Please specify.

LN558-559: A comparison between metazoan TP (based on d15N-Glu) and protozoan TA (based on d15N-Ala) will provide stronger support for this statement.

LN567: There is no direct evidence for copepods in this study.

LN574-576: Instead, state that “… were more enriched below 20 m depth compared to … at the surface”

LN578: Replace “enrichment of the heavier N isotopes” by “enrichment of the N isotopes”.

LN581: Trophic transfers do not enrich d15N of source AAs; your previous interpretation only mentioned degradation. Move LN584-587 to LN580 to support your conclusion.

LN592: Remove “to”.

LN591-594: The increased ∆Tr and the depleted d15N-Thr indicate that the higher ∑V could be due to more intense trophic transfers. Therefore, it cannot support the statement of higher microbial activity. Please revise your conclusion."

LN599: What findings? Please specify. The authors should elaborate on how evidence of trophic transfers can be distinguished from microbial degradation and identify parameters that are affected by both processes synchronically. Additionally, please detail how microbial communities in OMZs differ from those in oxygenated waters. Currently, these statements are too ambiguous.

LN615: I disagree. The d15N-Phe is lower in deeper waters but still higher compared to surface values, and the d15N-Thr is depleted, indicating heterotrophic biomass.

LN623: ∆Tr and ∑V do not imply respiration intensity or the amount and quality of exported material.

LN628: This is just a hypothesis, so be cautious with your interpretation.

LN635: “Mesopelagic water columns” from where? Please specify the location.

LN636: Are ∆Tr and ∑V changes correlated with the POC fluxes? Are the differences significant between the two bays?

LN644: The difference in d15N-Thr offsets between the two bays is a significant finding and deserves an elaborated paragraph, together with d15N-Ala. It should be mentioned alongside the discussion on ∆Tr and ∑V.

LN648: Please specify what is “This heterogeneity”.

LN650: The wording of this sentence is awkward. Please revise.

LN652: Replace “although… were observed” by ““despite”

LN656: Remove “to us”

LN653-654: This sentence is confusing. It should clarify the specific parameters that are similar (oxygen, temperature, salinity, pH, and chlorophyll a) despite the different vertical CSIA-AA profiles between the two bays. Was chlorophyll a compared? Productivity may also be a factor to consider.

LN659: This paragraph details how potential topography features in the two bays affect OM cycling. The authors effectively challenge the paradigm that water column heterogeneity alone affects OM cycling, which is a valid point. This should be mentioned in the abstract and conclusion. It could also be beneficial to discuss how coastal environments can differ significantly from pelagic environments in terms of physical, chemical, and biological interactions.

LN676: Please remove phrases like 'it was observed that' if you are already citing figures or references.

Conclusion: It could benefit from more detailed implications and outlook. Consider elaborating on how future studies might improve by including additional parameters, innovatively testing hypothese, and addressing the implications for global and local-scale climate change.

Reviewer #2: The manuscript is interesting and reports data on variables that are innovative for ecological research. The text is very long. Especially in the results section it should be shortened and simplified, modifying the figures and adding some other figure or tables to help the reader to understand which of the trends-variables-values are pivotal for the discussion. The discussion should be shortened, especially avoiding long hydrographic descriptions, and a little reorganised.

Detailed comments below.

Introduction

Lines 38-39: POM includes living organisms such as phytoplankton (see line 45), I would turn the sentence saying that inside POM a large fraction can be detrital

Lines 90-91: Does it mean that the circulation previously described is inverted? Maybe the main hydrodynamics can be provided in Fig. 1 to help the reader.

M&M

Line 108: I never found the word “chlorine” for the sum of chlorophyll-a and phaeopigment, are you sure?

Line 112: can you signal the special stations on Fig. 1 to help the reader? What is chlorine index? You describe the values in the results but do not explain how you calculated it.

Line 116: not in station E2BM , can you specify the variables studied in the trap material as you did for the water?

Line 135: this is the method generally used for dissolved C, I suppose you did not filter the water and therefore you obtained, as you say, the total C. But how did you measure the particulate fraction?

Line 190: can you add a simple sentence related to the interpretation of ΣV values?

Paragraph 2.7 should be placed before paragraph 2.6, since it identifies the trophic AA.

Line 224: correct AUO.

Results

Can you use the same scale for all the figures related to the same variable when comparing the two bays? Sometimes it is misleading to consider only the trends.

Line 263: in general, the trends of the variables are similar in the two bays, but some differences can be observed for instance at the surface. In the Discussion (line 669) you emphasise these differences, in contradiction with the Results. Horizontal plots for T at surface and 5 m depth (the bottom of the surface layer) could give a more detailed description of the variability of the hydrological features within each bay and help to discuss any differences you find in the two transects sampled for isotopic analyses.

Line 264: In Fig. 2, I can’t find the waters mentioned in the text. The AESS box doesn’t contain any observations, the other two boxes (ASSA-ASST) are not described-cited in the text. The Brunt Väisälä Frequency is not described, although it is reported in Fig. 6C in the Discussion, you should place this figure here (the figure is quite difficult to understand, I would divide the two areas to have larger plots with clear position of fronts and, if available, the direction of the currents).

Line 274: San Jorge Bay is Antofagasta Bay; you should cite this in the Study site paragraph.

Lines from 285: please add in the figures a small map of the stations, to help the reader to immediately find where the cited stations are located. For instance, the chlorophyll-a value of station E1BA at 20 m depth is unique, maybe because the station is placed next to the coast.

Results, as often occurs due to the number of available data, are not easy to follow. The supplementary figures are “overcrowded” , for chlorophyll-a a horizontal plot focused on 5 m depth could help (the plot in the text and maintaining the S figure?). For nutrients, while NH4 and NO2 show scattered values, NO3 and PO4 seem to follow quite regular trends with dept, but maybe at surface they have a distribution that is reflected by primary biomass (or hydrography?). Are there any significant relationships between nutrients and chlorophyll-a (or the chlorine index and TOC)? In that case, you can represent them as you did in Fig. S6 for environmental data, or in a Table. Focusing the description of the results on these observations would help the reader more than knowing the highest and the lowest values.

Line 329: I think this is suspended, not sinking. Also here to follow the text is difficult, can you refer briefly to table 1? North and south for the two transects are not only related to latitude, but to proximity to the coast or to the head of the bay, that is the opposite in the two bays. You use this way to describe the stations-data in the discussion for physical variables, maybe it is better to use it everywhere.

Why did you choose to report in Table1 the values related to Phe and Thr, both source AA, and not an example of trophic AA?

Line 390 (and maybe elsewhere): check the figure number, ΣV is panel C in figures 4 and 5.

Fig.3 is complex, in the text you place together the description of trophic AA and of source AA, a figure with average values of the two kinds of AA would be easier to understand. In the supplementary material you can add a Table for the data.

Discussion

The first part is related to NO3, I would change the title of the paragraph.

Lines 426-444 : a division of the text in sub-paragraphs would help the reader. I would start with the denitrification influence on the entire water column (upwelling of water from OMZ) (lines 426-433), thus I would focus on the surface layer, where phytoplankton can have a role (433-436 + 438-444).

Line 445: Until now you have compared your data with those coming from other areas, why do you say: another explanation…?

Line 450-454: please simplify the sentence and place the information at the beginning of the Discussion together with the comparisons with other areas.

Lines 460- 470: I would reduce the length of the paragraph and add the information at the beginning, where the relevance of nitrification in the entire water column is underlined.

Lines 471-493 : I would strongly reduce the general description of the hydrodynamics and focus on the stratification (Brunt Vaisala may help).

From line 494 to 520 the POM is concerned; I would start a new paragraph with a new title.

Line 521-539 and following: why contrarily? Is “it” referred to the marine origin? Is this the place for ΣV? I would start here the new paragraph on microbial reworking and trophic transfer (try to simplify, remove lines 543-553).

Line 559: zooplanktonic means microbial? That is, heterotrophic? This would better link with the previous discussion on enzymatic activity. Are you sure that your data support the zooplanktonic presence in the area? This component is important, and it has the functions you indicated, but I’m not sure that your data strongly suggest this. ΔTr lower than 1 should mean that the consumers are not dominant. You should explain this point better.

Line 564: surface layer is better than shallower

Lines 574-587: trophic transfer means also zooplanktonic reworking of POM? Does zooplankton live in the deep water column of the bays? I’m lost: at surface lower ΔTr means zooplankton and in the depth higher ΔTr means microbes?

Line 587: is it NH4 or NO3?

Line 588: POC decrease with depth is not surprising, given the strong pycnocline that maintain the OM produced at surface in the first metres.

Line 628: the enzymatic activity is a function of the ecosystem that can be found everywhere, generally it depends on substrate availability.

Line 639: is the difference significant?

Line 643: why Thr behaves differently?

In this latter part of the Discussion, I would avoid repetition of the results and hypotheses that cannot be confirmed by the data, and therefore Line 658 conclusion.

Lines 659-684 : a summary of this information should be placed before, when you discuss vertical and horizontal trends.

6. PLOS authors have the option to publish the peer review history of their article (what does this mean? ). If published, this will include your full peer review and any attached files.

**Do you want your identity to be public for this peer review?** For information about this choice, including consent withdrawal, please see our Privacy Policy .

Reviewer #1: No

Reviewer #2: No

---

## [Author Response · Author response to Decision Letter 1]

22 Jan 2025

I. Academic Editor

Response: Done

Response: Added in lines 152-154 of the revised manuscript.

Response: Edited and removed from the “Acknowledgments” section.

4. Please state what role the funders took in the study.

5. We note that your Data Availability Statement is currently as follows: All relevant data are within the manuscript and its Supporting Information files.

Response: We have provided all metadata used in the manuscript as "Supporting Information files."

We have included the statement, "The funders had no role in study design, data collection and analysis, decision to publish, or preparation of the manuscript," in the new cover letter.

II. Reviewer # 1

General Comments

Calculation of trophic transfer is outdated

Response: As suggested by Reviewer 1, we have calculated TP Metazoan and TP Protozoan. Please refer to lines 274 and 278 in the revised manuscript for details

The grouping of source and trophic AAs is incorrect

Response: We have corrected it. Please refer to Line 261 in the revised manuscript.

Phe-normalized δ15N values of Ala and Thr are not calculated

Response: These calculations were performed, and the results are discussed in the revised manuscript. Please refer to lines 304-310 for details.

δ15N -THAA is not “weight-averaged”

Response: In the revised manuscript, we calculated and presented the " mole percent weighted sum” of δ15N and δ13C -THAA" (Lines 295-302). These values were retained and discussed in the text, emphasizing their role as a "proxy" to estimate the bulk δ15N and δ13C of POM. The obtained values were compared with previously obtained δ15N PON and δ13C POC values in coastal and epipelagic waters of Chile.

5) Additionally, the wording and phrasing need to be more specific, concise, and scientifically rigorous. There are instances of redundant wording, such as 'heavier δ15N enrichment,' and overly long sentences. The overall structure of the manuscript could also benefit from improvement.

Response: We have addressed these concerns by improving the writing throughout the entire manuscript.

Specific Comments

Abstract: When summarizing the findings of δ15N-NO3-, δ15N -Phe, ∆Tr and ∑V, consider reporting the actual numerical values instead of providing a general description that these parameters led to certain conclusions

Response: We have included numerical values of ∑V, TP Metazoan and TP Protozoan in the abstract of the revised manuscript.

LN14: The authors started the abstract with a general description of pelagic POM which does not serve as a strong motivation for this study since pelagic POM has been widely studied. Instead, highlighting POM processing in the productive and oxygen-deficient upwelling systems would be more compelling for readers. Consider emphasizing the uniqueness of the study area rather than the general importance of POM.

Response: In the revised manuscript, we begin with a line highlighting the significance of upwelling bays and their role in primary production. Please refer to the abstract.

LN16: Replace “chemical” by “characterizable”. With increasing depth, there is an increase in uncharacterizable POM in degraded organic matter, which sometimes constitutes the largest portion overall, whereas amino acids account for the largest portion in the characterizable portion.

Response: This paragraph has been removed from the revised manuscript.

LN21-22: The study objectives need to be more specific than simply “gaining insight”. Consider detailing objectives such as testing the differences between the two bays and/or examining the correlations between heterotrophic processing and oceanographic conditions

Response: The main objectives were rethought and included as suggested by Reviewer #1 (Lines 86-93 of the revised manuscript).

LN23-24: The ranges of δ15N -AA are neither important nor informative results and do not need to be reported in the abstract. I suggest reporting the average δ15N of source and trophic AAs (groupings to be revised – see LN211)

Response: This paragraph has been removed from the revised manuscript. Additionally, we calculated and presented the average δ15N of source and trophic AAs in the revised manuscript (Lines 403-428).

LN30: Remove “at” or “through”

Response: Removed

LN34: Replace “this” by “the northern Chilian”

Response: Removed

LN45: Replace “largest” by “largest characterizable

Response: Replaced in the revised manuscript.

LN54: The explanation of why δ15N -AA reflects trophic transfers and degradation is not clear. The grouping of source and trophic AAs should be described here.

Response: Addressed and rewritten in the revised manuscript (Lines 71-78).

Methods: The overall structure of this section is disorganized. It should follow a logical order: sampling, processing, analysis, and calculations/statistics. Consider starting with the sampling of water samples and the deployment and collection of sediment traps, followed by their processing and analyses. Providing a summary paragraph detailing the parameters measured for water samples and sediment traps respectively would enhance clarity for readers.

Response: We have reorganized and rewritten this section. Please refer to lines 96-324 in the revised manuscript

LN111: Replace “three special stations were established” by “three stations were established for stable nitrogen isotope analysis”

Response: Replaced in the revised manuscript (Lines 138-139).

LN115: Replace “These subsamples” by “Water samples for δ15N N-NO3- analysis”

Response: Replaced. Line 145 in the revised manuscript.

LN117: Remove “Additionally, … ocean floor” and insert a paragraph/section of sediment trap deployment here

Response: Removed, and a paragraph/section on sediment trap deployment was added in the revised manuscript (Lines 156-164).

LN143: The term “all samples” is confusing because the authors mentioned that amino acid δ15N was measured only in samples from three stations. Please specify which samples from which stations were included in the analysis.

Response: Rewritten in the revised manuscript.

LN157: The reproducibility of C and N of amino acid stable isotopes are typically different. It would be helpful to report specific reproducibility values for C and N separately.

Response: Included in the revised manuscript. Please refer to lines 244 and 247 in the revised manuscript.

LN178: Standards for nitrate δ15N measurements should be reported

Response: Reported in the revised manuscript (Lines 224-227).

Sect. 2.6: This section requires major revision. Firstly, the calculation of trophic transfer appears outdated, as newer equations based on d15N of Phe and Glu (for metazoan foodwebs, Chikaraishi et al., 2009) or Ala (for protozoan foodwebs, Décima et al., 2017) and improved β values and trophic enrichment factors from meta-analyses have been reported. It is advisable to recalculate trophic positions for metazoans (using Glu/Phe) and protozoans (using Ala/Phe), especially since the authors suggest that POM transfer is predominantly facilitated by zooplankton. Secondly, the authors should also calculate Phe-normalized Ala and Thr and compare their data with previously published end-members (zooplankton, fecal pellets, degraded OM, and phytoplankton). It would be beneficial to apply a Bayesian mixing model using these end-members to estimate the composition of OM. This approach would strengthen your interpretation regarding the degree of trophic transfer and microbial reworking.

Response: We have recalculated trophic positions for metazoan and protozoan using Glu/Phe and Ala/Phe, respectively (Lines 274 and 278 in the revised manuscript). Additionally, we calculated Phe-normalized δ15N Ala and Thr (Lines 304-310) to compare with Phe-normalized δ15N Ala and Thr of organic matter end-members previously published in the literature and used in Bayesian mixing model analysis (Lines 325-330). All descriptions and discussions have been included in the revised manuscript.

LN202: Remove the first “-”. Insert “δ13C and” between “the” and “δ15N”. δ13C and δ15N values of THAA are typically weight-averaged due to variations in the concentration of individual amino acids in POM, which affects the overall isotopic value. Authors should recalculate δ13C- and δ15N-THAA values based on amino acid concentration profiles. Additionally, THAA δ13C and δ15N may not accurately represent bulk δ13C and δ15N, especially for C. If feasible, authors should measure bulk stable isotopes separately and include these measurements in the study. Otherwise, all interpretations assuming that THAA represents “bulk” values should be removed from the manuscript.

Response: In the revised manuscript, we calculated and presented the "mole percent weighted sum" of δ15N and δ13C -THAA (Lines 307-314). These values were retained and discussed in the text, emphasizing their role as a "proxy" to “estimate” the bulk δ15N and δ13C of POM. The obtained values were compared with previously obtained δ15N PON and δ13C POC values in coastal and epipelagic waters of Chile to enhance the consistency of our discussion and conclusions.

LN211: Remove “Gly” and “Thr”. In more recent studies, Gly and Thr are not considered “source AAs” anymore, especially for Thr that gets depleted with trophic transfer. Most studies do not assign Gly and Thr to specific groupings

Response: Gly and Ser were removed from the source AA grouping in the revised manuscript.

LN224: Replace “AUO” by “AOU”

Response: Replaced in the revised manuscript.

LN256: Please specify “significant differences” in which parameters and between which groups or conditions.

Response: It has been specified in the revised manuscript (Lines 312-316).

Results:

LN312: The authors did not interpret chlorine index in the discussion. Consider either removing this section or moving it to the supplementary materials.

Table 1: Since the authors mentioned that the reproducibility of δ13C and δ15N measurements is better than 1‰, rounding the δ13C and δ15N values to integers does not provide sufficient precision. Most studies report stable isotope values to the tenth place

Response: The chlorin index was removed from the manuscript. All isotopic data presented in graphs and tables are now shown to the tenth place in the revised manuscript.

LN329-331: Given the wide ranges of δ15N -AA from 5 to 27‰ and 9 to 24‰, it's impossible for the standard deviations of the average values to be 0.1‰. Due to this variability, reporting the average values may not provide meaningful information

Response: Corrected in the revised manuscript.

LN331-332: This sentence is confusing – were the differences insignificant for all the AAs between the two bays? According to Table 2, there appear to be large differences (up to 3-4‰) in certain AAs between the two bays.

Response: Sentence was removed and rewritten in the revised manuscript. Please refer to lines 403-428.

LN334: Replace “most depleted” by “least enriched”.

Response: This phrase has been omitted from the revised manuscript.

LN335: Remove “Thr” from source AAs

Response: Removed

LN367: Sediment trap δ15N-AA data should be included in Figure 3, with the legend clearly indicating which data are from water columns and which are from traps

Response: The metadata for sediment traps δ15N-AA are provided separately in Table S3 of the Supporting Material. Values for derived δ15N-AA parameters for sinking POM are presented in Table 1 of the revised manuscript (Line 386).

LN371: Recalculate the average after removing Gly and Thr

Response: Average source-AA values were calculated excluding Gly and Thr, as described in line 272 in the revised manuscript.

LN377: Please refer to the comments for Sect 2.6

Response: This was addressed in previous responses, specifically concerning the observations in section 2.6

LN399-401: This point should be discussed in the discussion section

Response: Removed from revised manuscript.

Table 2: Please refer to the comments for Table 1. Standard deviations should be reported alongside the means

Response: Table 2 has been replaced by Table 1 in the revised manuscript. Values are presented as mean ± standard deviations (Line 386).

Discussion:

Sect 4.1: The authors spent a considerable amount of discussion speculating on how the distribution of δ15N-NO3- could be affected by various processes, despite lacking direct evidence from the current study. Please consider shortening this section and placing more focus on POM

Response: This section has been rewritten and summarized in the revised manuscript (Lines 484-507).

LN430: Please specify “other marine oxygen-depleted waters

Response: Sentence rewritten in the revised manuscript (Line 487-490).

LN436-438: The authors discussed how upwelling of 15N-enriched nitrogen and incomplete nitrate utilization may lead to the relatively high δ15N-NO3- compared to the Pacific average. However, I disagree that incomplete nitrate assimilation by phytoplankton is a major mechanism contributing to the enriched values since we do not observe a large discrepancy between surface δ15N-NO3- and δ15N -Phe (an indicator of phytoplankton biomass). The surface NO3- is already 15N -enriched as a result of upwelled regenerated nitrogen before being assimilated into phytoplankton.

Response: The paragraph has been removed and rewritten in the revised manuscript (Lines 491-502).

LN453-454: The δ15N-NO3- values in deep waters are highly variable between geographical locations, and comparison with the world average does not provide meaningful information. It would be more informative to compare these values with deep waters from adjacent locations or similar upwelling systems.

Response: The paragraph has been rewritten in the revised manuscript (Lines 485-490).

LN494: Start this sentence with POM values, stating that they are similar to δ15N-NO3-, as the focus here is on POM

Response: The sentence has been rewritten in the revised manuscript.

LN506-508: Please refer to comments regarding THAA and bulk δ15N values for

Response: This was addressed in previous responses, specifically concerning the observations in LN202.

LN202. Since THAA cannot accurately represent bulk organic material and the THAA measured in this study does not constitute a full suite of AAs, direct comparisons in δ15N should not be made

Response: This was addressed in previous responses, specifically concerning the observations in LN202.

LN521: The sentence starting with “Contrarily, it highlights the fact that” is confusing because it's unclear what 'it' refers to and why 'it' is considered contrary. Contrary to what?

Response: Paragraph removed from the revised manuscript

LN527-532: This sentence is overly lengthy.

Response. Sentence has been rewritten

LN527: Please provide depth distributions of δ15N -TrAA and Phe-normalized δ15N -TrAA to support your statement regarding synchronous enrichment of d15N

Response: This is illustrated in figure 4C and supplementary figure 5A.

LN530: External hydrolysis is a hypothesis; please be cautious with your interpretation

Response: The sentence has been rewritten in the revised manuscript (Lines 653-655).

LN537-539: Move the sentence to LN532 before “In oxic…

Response: The sentence has been relocated in the revised manuscript.

Sect 4.2: Please start the section with a summary sentence or paragraph. There is a lot of information in the section. Consider breaking it into two subsections: one focusing on vertical profiles and the other on spatial heterogeneity

Response: In the revised manuscript, the discussion section is divided as follows: 4.1. Dynamics of δ15N-NO3- in Mejillones and Antofagasta Bays; 4.

---

## [Decision Letter · Decision Letter 1]

9 Mar 2025

PONE-D-24-25324R1ORIGIN, TROPHIC TRANSFER AND RECYCLING OF PARTICULATE ORGANIC MATTER IN THE WATERS OF TWO UPWELLING BAYS OF HUMBOLDT CURRENT SYSTEM: INSIGHTS FROM COMPOUND-SPECIFIC ISOTOPIC COMPOSITIONS OF AMINO ACIDSPLOS ONE

Dear Dr. Srain,

Thank you for submitting your manuscript to PLOS ONE. After careful consideration, we feel that it has merit but does not fully meet PLOS ONE’s publication criteria as it currently stands. Therefore, we invite you to submit a revised version of the manuscript that addresses the points raised during the review process.

We look forward to receiving your revised manuscript.

Kind regards,

Vitor Hugo Rodrigues Paiva, Ph.D.

Academic Editor

PLOS ONE

Journal Requirements:

Reviewers' comments:

Reviewer's Responses to Questions

**Comments to the Author**

1. If the authors have adequately addressed your comments raised in a previous round of review and you feel that this manuscript is now acceptable for publication, you may indicate that here to bypass the “Comments to the Author” section, enter your conflict of interest statement in the “Confidential to Editor” section, and submit your "Accept" recommendation.

Reviewer #1: All comments have been addressed

Reviewer #3: All comments have been addressed

2. Is the manuscript technically sound, and do the data support the conclusions?

Reviewer #1: (No Response)

Reviewer #3: Yes

3. Has the statistical analysis been performed appropriately and rigorously? 

Reviewer #1: (No Response)

Reviewer #3: Yes

4. Have the authors made all data underlying the findings in their manuscript fully available?

Reviewer #1: (No Response)

Reviewer #3: Yes

5. Is the manuscript presented in an intelligible fashion and written in standard English?

Reviewer #1: (No Response)

Reviewer #3: Yes

6. Review Comments to the Author

Reviewer #1: (No Response)

Reviewer #3: The authors conducted a good comparative analysis of the POM dynamics in two upwelling bays and found significant differences in POM dynamics between the two bays, mainly attributed to different biological, hydrological, and topographical characteristics. Overall, authors addressed all the comments from reviewers and editors appropriately, I recommend accepting this paper after the following minor revisions.

1. Lines 22-29. The presentation of the abstract is still somewhat vague. The authors first presented the overall characteristics of POM sources and isotopic differences, and finally merged all influencing factors together. Authors should explain each result separately, such as the reasons for the differences in POM sources and isotope values.

2. Line 394. The presentation of the "P" in statistical analysis should be italicized. Please unify the entire text.

3. Lines 662-663. N* has already been defined in the previous text, there is no need to repeat it here.

7. PLOS authors have the option to publish the peer review history of their article (what does this mean? ). If published, this will include your full peer review and any attached files.

**Do you want your identity to be public for this peer review?** For information about this choice, including consent withdrawal, please see our Privacy Policy .

Reviewer #1: No

Reviewer #3: No

---

## [Author Response · Author response to Decision Letter 2]

20 Mar 2025

We have addressed all the corrections suggested by the editor and Reviewer #3.

---

## [Editor Report · Decision Letter 2]

31 Mar 2025

ORIGIN, TROPHIC TRANSFER AND RECYCLING OF PARTICULATE ORGANIC MATTER IN TWO UPWELLING BAYS OF HUMBOLDT CURRENT SYSTEM: INSIGHTS FROM COMPOUND-SPECIFIC ISOTOPIC COMPOSITIONS OF AMINO ACIDS

PONE-D-24-25324R2

Dear Dr. Srain,

We’re pleased to inform you that your manuscript has been judged scientifically suitable for publication and will be formally accepted for publication once it meets all outstanding technical requirements.

Kind regards,

Vitor Hugo Rodrigues Paiva, Ph.D.

Academic Editor

PLOS ONE
---

## [Editor Report · Acceptance letter]

PONE-D-24-25324R2

PLOS ONE

Dear Dr. Srain,

I'm pleased to inform you that your manuscript has been deemed suitable for publication in PLOS ONE. Congratulations! Your manuscript is now being handed over to our production team.

Kind regards,

on behalf of

Dr. Vitor Hugo Rodrigues Paiva

Academic Editor

PLOS ONE